# Continuous Control with Action Quantization from Demonstrations

## Abstract

In Reinforcement Learning (RL), discrete actions, as opposed to continuous actions, result in less complex exploration problems and the immediate computation of the maximum of the action-value function which is central to dynamic programming-based methods. In this paper, we propose a novel method: Action Quantization from Demonstrations (AQuaDem) to learn a discretization of continuous action spaces by leveraging the priors of demonstrations. This dramatically reduces the exploration problem, since the actions faced by the agent not only are in a finite number but also are plausible in light of the demonstrator's behavior. By discretizing the action space we can apply any discrete action deep RL algorithm to the continuous control problem. We evaluate the proposed method on three different setups: RL with demonstrations, RL with play data –demonstrations of a human playing in an environment but not solving any specific task– and Imitation Learning. For all three setups, we only consider human data, which is more challenging than synthetic data. We found that AQuaDem consistently outperforms state-of-the-art continuous control methods, both in terms of performance and sample efficiency on a variety of hard manipulation tasks.

## 1 Introduction

With several successes on highly challenging tasks including strategy games such as Go (Silver et al., 2016), StarCraft (Vinyals et al., 2019) or Dota 2 (Berner et al., 2019) as well as robotic manipulation (Andrychowicz et al., 2020), Reinforcement Learning (RL) holds a tremendous potential for solving sequential decision making problems. RL relies on Markov Decision Processes (MDP) (Puterman, 2014) as its cornerstone, a general framework under which vastly different problems can be casted.

There is a clear separation in the class of MDPs between the finite discrete action setup, where an agent faces a finite number of possible actions, and the continuous action setup, where an agent faces an infinite number of actions. When the number of actions is small, the former is arguably simpler, since exploration is more manageable with a finite, reasonable number of actions, and computing the maximum of the action-value function is straightforward (and implicitly defines a greedily-improved policy). In the continuous action setup, the parametrized policy either directly optimizes the expected value function that is estimated through Monte Carlo rollouts (Williams, 1992), which makes it demanding in interactions with the environment, or optimizes a parametrized state-action value function (Konda and Tsitsiklis, 2000) hence introducing additional sources of approximations.

Therefore, a workaround consists in turning a continuous control problem into a discrete one. The simplest approach is to naively (*e.g.* uniformly) discretize the action space, an idea which dates back to the "bang-bang" controller (Bushaw, 1952). However, such a discretization scheme suffers from the curse of dimensionality. Various methods have addressed this limitation by making the strong assumption of independence (Tavakoli et al., 2018; Tang and Agrawal, 2020; Andrychowicz et al., 2020) or of causal dependence (Metz et al., 2017; Vinyals et al., 2019; Sakryukin et al., 2020; Tavakoli et al., 2021) between the action dimensions which are typically complex and task-specific (e.g. autoregressive policies, pointer networks based architectures).

In this work, we introduce a novel approach leveraging the prior of human demonstrations for reducing a continuous action spaces to a discrete set of meaningful actions. The proposed method does not suffer from the curse of dimensionality and does not require any task specific assumption.

Demonstrations typically consist of transitions experienced by a human in the targeted environment, performing the task at hand or not. They are of particular interest in cases where the reward function is hard to define (Russell, 1998; Ng et al., 1999), to facilitate exploration (Salimans and Chen, 2018; Nair et al., 2018) or to build behavioral priors (Singh et al., 2020).

We thus propose Action Quantization from Demonstrations, or AQuaDem, a novel paradigm where we learn a state dependent discretization of a continuous action space using demonstrations, enabling the use of discrete-action deep RL methods by virtue of this learned discretization. We formalize this paradigm, provide a neural implementation and analyze it through visualizations in simple grid worlds. We empirically evaluate this discretization strategy on three downstream task setups: Reinforcement Learning with demonstrations, Reinforcement Learning with play data, and Imitation Learning. We test the resulting methods on robotics tasks and show that they outperform state-of-the-art continuous control methods both in terms of sample-efficiency and performance on every setup.

## 2 PRELIMINARIES

**Markov Decision Process.** We model the sequential decision making problem as a Markov Decision Process (MDP) (Puterman, 2014; Sutton and Barto, 2018). An MDP is a tuple $(\mathcal{S}, \mathcal{A}, \mathcal{P}, r, \gamma, \rho_0)$, where $\mathcal{S}$ is the state space, $\mathcal{A}$ is the action space, $\mathcal{P}$ is the transition kernel, $r$ is the expected reward function, $\gamma$ the discount factor and $\rho_0$ the initial state distribution. Throughout the paper, we distinguish discrete action spaces, which simply amount to a set $\{1, \ldots, K\}$, from continuous action spaces which consist in an interval of $\mathbb{R}^d$ where $d$ is the dimensionality of the action space. A stationary policy $\pi$ is a mapping from states to distributions over actions. The value function $V^\pi$ of a policy $\pi$ is defined as the expected discounted cumulative reward from starting in a particular state and acting according to $\pi$: $V^\pi(s) = \mathbb{E}\left[\sum_{t=0}^\infty \gamma^t r(s_t, a_t) | s_0 = s, a_t \sim \pi(s_t), s_{t+1} \sim \mathcal{P}(s_t, a_t))\right]$. An optimal policy $\pi^*$ maximizes the value function $V^{\pi^*}$ for all states. The action-value function $Q^\pi$ is defined as the expected discounted cumulative reward from starting in a particular state, taking an action and then acting according to $\pi$: $Q^\pi(s, a) = r(s, a) + \gamma \mathbb{E}\left[V^\pi(s') | s' \sim \mathcal{P}(s, a)\right]$.

**Value-based RL.** The Bellman (1957) operator $\mathcal{T}$ connects an action-value function $Q$ for the state-action pair $(s, a)$ to the action-value function in the subsequent state $s'$: $\mathcal{T}^\pi(Q)(s, a) := r(s, a) + \gamma \mathbb{E}\left[Q(s', a') | a' \sim \pi(s), s' \sim \mathcal{P}(s, a)\right]$. Value Iteration (VI) (Bertsekas, 2000) is the basis for methods using the Bellman equation to derive algorithms estimating the optimal policy $\pi^*$. The prototypical example is the $Q$-learning algorithm (Watkins and Dayan, 1992), which is the basis of *e.g.* DQN (Mnih et al., 2015), and consists in the repeated application of a stochastic approximation of the Bellman operator $Q(s, a) := r(s, a) + \gamma \max_{a'} Q(s', a')$, where $(s, a, s')$ is a transition sampled from the MDP. The $Q$-learning algorithm exemplifies two desirable traits of VI-inspired methods in discrete action spaces that are **1)** bootstrapping: the current $Q$-value estimate at the next state $s'$ is used to compute a finer estimate of the $Q$-value at state $s$, and **2)** the exact derivation of the maximum $Q$-value at a given state. For continuous action spaces, state-of-the-art methods (Haarnoja et al., 2018; Fujimoto et al., 2018) are also fundamentally close to a VI scheme, as they rely on Bellman consistency, with the difference being that the argument maximizing the $Q$-value, in other words the parametrized policy, is approximate.

**Demonstration data.** Additional data consisting of transitions from an agent may be available. These demonstrations may contain the reward information or not. In the context of Imitation Learning (Pomerleau, 1991; Ng et al., 1999; 2000; Ziebart et al., 2008), the assumption is that the agent generating the demonstration data is near-optimal and that demonstration rewards are not provided. The objective is then to match the distribution of the agent with the one of the expert. In the context of Reinforcement Learning with demonstrations (RLfD) (Hester et al., 2018; Vecerik et al., 2017), demonstration rewards are provided. They are typically used in the form of auxiliary objectives together with a standard learning agent whose goal is to maximize the environment reward. In the context of Reinforcement Learning with play (Lynch et al., 2020), demonstration rewards are not provided as play data is typically not task-specific.

Demonstration data can come from various sources, although a common assumption is that it is generated by a single, unimodal Markovian policy. However, most of available data comes from agents that do not fulfill this condition. In particular, for human data, and even more so when coming from several individuals, the behavior generating the episodes may not be unimodal nor Markovian.

# 3 METHOD

In this section, we introduce the AQuaDem framework and a practical neural network implementation together with an accompanying objective function. We provide a series of visualizations to study the candidate actions learned with AQuaDem in gridworld experiments.

**Step 1** (offline) Learn state-conditioned quantization.    **Step 2** (online) Run discrete RL on quantized actions.

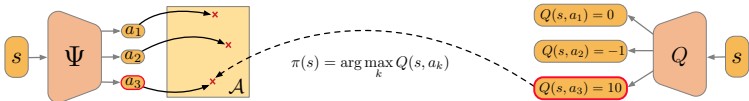

Visualization of the AQuaDem framework (offline) with a downstream algorithm (online).

## 3.1 AQUADEM: ACTION QUANTIZATION FROM DEMONSTRATIONS

Our objective is to reduce a continuous control problem to a discrete action one, on which we can apply discrete-action RL methods. Using demonstrations, we wish to assign to each state $s \in \mathcal{S}$ a set of $K$ candidate actions from $\mathcal{A}$. The resulting action space is therefore a discrete finite set of $K$ state-conditioned vectors. In a given state $s \in \mathcal{S}$, picking action $k \in \{1, \ldots, K\}$ stands for picking the $k^{\text{th}}$ candidate action for that particular state. The AQuaDem framework refers to the discretization of the action space, and the resulting discrete action algorithms used with AQuaDem on continuous control tasks are detailed in Section 4. We propose to learn the discrete action space through a modified version of the Behavioral Cloning (BC) (Pomerleau, 1991) reconstruction loss that captures the multimodality of demonstrations. Indeed the typical BC implementation consists in building a deterministic mapping between states and actions $\Phi : \mathcal{S} \mapsto \mathcal{A}$. But in practice, and in particular when the demonstrator is human, the demonstrator can take multiple actions in a given state (we say that its behavior is *multimodal*) which are all good candidates for AQuaDem. We thus learn a mapping $\Psi : \mathcal{S} \mapsto \mathcal{A}^K$ from states to a set of $K$ candidate actions and optimize a reconstruction loss based on a soft minimum between the candidate actions and the demonstrated action.

Suppose we have a dataset of expert demonstrations $\mathcal{D} = \{(s_i, a_i)\}_{1:n}$. In the continuous action setting, the vanilla BC approach consists in finding a parametrized function $f_\Phi$ that minimizes the reconstruction error between predicted actions and actions in the dataset $\mathcal{D}$. To ease notations, we will conflate the function $f_\Phi$ with its parameters $\Phi$ and simply note it $\Phi : \mathcal{S} \mapsto \mathcal{A}$. The objective is thus to minimize: $\min_\Phi \mathbb{E}_{s,a\sim\mathcal{D}} \|\Phi(s) - a\|^2$. Instead, we propose to learn a set of $K$ actions $\Psi_k(s)$ for each state by minimizing the following loss:

$$\min_\Psi \mathbb{E}_{s,a\sim\mathcal{D}} \Big[ -\log \Big( \sum_{k=1}^{K} \exp \big(\frac{-\|\Psi_k(s) - a\|^2}{T}\big)\Big)\Big], \tag{1}$$

where the temperature $T$ is a hyperparameter. Equation (1) corresponds to minimizing a soft-minimum between the candidates actions $\Psi_1(s), \ldots, \Psi_K(s)$ and the demonstrated action $a$. Note that with $K = 1$, this is exactly the BC loss. The larger the temperature $T$ is, the more the loss imposes all candidate actions to be close to the demonstrated action $a$ thus reducing to the BC loss. The lower the temperature $T$ is, the more the loss only imposes a single candidate action to be close to the demonstrated action $a$. We provide empirical evidence of this phenomenon in Section 3.2. Equation (1) is also interpretable in the context of Gaussian mixture models (see Appendix A). The $\Psi$ function enables us to define a new MDP where the continuous action space is replaced by a discrete action space of size $K$ corresponding to the $K$ action candidates returned by $\Psi$ at each state.

## 3.2 VISUALIZATION

In this section, we analyze the actions learned through the AQuaDem framework, in a toy grid world environment. We introduce a continuous action grid world with demonstrations in Figure 1.

We define a neural network $\Psi$ and optimize its parameters by minimizing the objective function defined in Equation (1) (implementation details can be found in Appendix D.1). We display the resulting candidate actions across the state space in Figure 2. As each color of the arrows depicts a

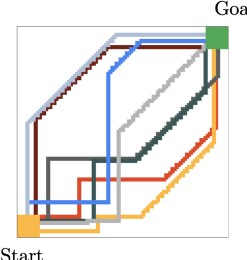

Figure 1: Grid world environment where the start state is in the bottom left, and the goal state is in the top right. Actions are continuous (2-dimensional), and give the direction in which the agent take a step. These steps are normalized by the environment to have fixed L2 norm. The stochastic demonstrator moves either right or up in the bottom left of the environment then moves diagonally until reaching the edge of the grid, and goes either up or right to reach the target. The demonstrations are represented in the different colors.

single head of the $\Psi$ network, we observe that the action candidates are *smooth*: action candidates slowly vary as the state vary, which prevents to have inconsistent action candidates in nearby states. Note that BC actions tend to be *diagonal* even in the bottom left part of the action space, where the demonstrator only takes horizontal or vertical actions. On the contrary, the action candidates learned by AQuaDem include the actions taken by the demonstrator conditioned on the states. Remark that in the case of $K = 2$, the action *right* is learned independently of the state position (middle plot in Figure 2) although it is only executed in a subspace of the action space. In the case of $K = 3$, actions are completely state-independent. In non-trivial tasks, the state dependence induced by the AQuaDem framework is essential, as we show in the ablation study in Appendix C and in the analysis of the actions learned in a more realistic setup in Appendix B.

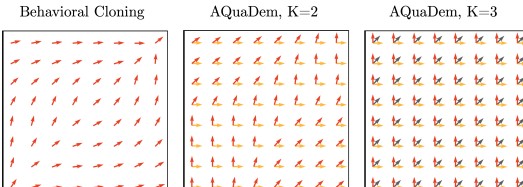

Figure 2: Visualisation of the action mapping learned by BC and the candidate actions learned with AQuaDem for $K = 2$ and $K = 3$ and $T = 0.01$. Each color represents a head of the $\Psi$ network.

**Influence of the temperature.** The temperature controls the degree of smoothness of the soft-minimum defined in Equation (1). We show that with larger temperatures, the soft-minimum converges to the average which is well represented in Figure 3 rightmost plot where the profile of AQuaDem's action candidates conflate with actions learned by BC. With lower temperatures, the actions taken by the demonstrator are recovered, but if the temperature is too low ($T = 0.001$), some actions that are not taken by the demonstrator might appear as candidates (blue arrows in the leftmost figure). This occurs because the soft minimum converges to a hard minimum with lower temperatures meaning that as long as one candidate is close enough to the demonstrated action, the other candidates can be arbitrarily far off. In this work, we treat the temperature as a hyperparameter, although a natural direction for future work is to aggregate actions learned for different temperatures.

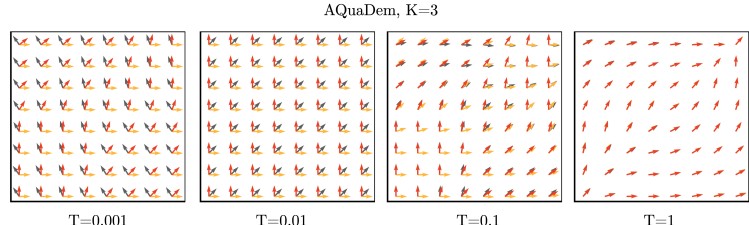

Figure 3: Influence of the temperature on resulting candidate actions learned with AQuaDem.

## 3.3 DISCUSSION

**On *losing* the optimal policy.** In any form of discretization scheme, the resulting class of policies might not include the optimal policy of the original MDP. In the case of AQuaDem, this mainly depends on the quality of the demonstrations. For standard continuous control methods, the parametrization of the policy also constrains the space of possible policies, potentially not including the optimal one. This is a lesser problem since policies tend to be represented with functions with universal approximation capabilities. Nevertheless, for most continuous control methods, the policy improvement step is approximate, while in the case of AQuaDem it is exact, since it amounts to selecting the argmax of the $Q$-values.

**On the multimodality of demonstrations.** The multimodality of demonstrations enables us to define multiple plausible actions for the agent to take in a given state, guided by the priors of the demonstrations. We argue that the assumption of multimodality of the demonstrator should actually be systematic (Mandlekar et al., 2021). Indeed, the demonstrator can be *e.g.* non-Markovian, optimizing for something different than a reward function like curiosity (Barto et al., 2013), or they can be in the process of learning how to interact with the environment. When demonstrations are gathered from multiple demonstrators, this naturally leads to multiple modalities in the demonstrations. And even in the case where the demonstrator is optimal, multiple actions might be equally good (*e.g.* in navigation tasks). Finally, the demonstrator can interact with an environment without any task specific intent, which we refer to as *play* (Lynch et al., 2020) and also induces a multimodal behavior.

## 4 EXPERIMENTS

In this section, we evaluate the AQuaDem framework on three different downstream tasks setups: RL with demonstrations, RL with play data and Imitation Learning. For all experiments, we detail the networks architectures, hyperparameters search, and training procedures in the Appendix.

### 4.1 REINFORCEMENT LEARNING WITH DEMONSTRATIONS

**Setup.** In the Reinforcement Learning with demonstrations setup (RLfD), the environment of interest comes with a reward function and demonstrations (which include the reward), and the goal is to learn a policy that maximizes the expected return. This setup is particularly interesting for sparse reward tasks, where the reward function is easy to define (say reaching a goal state) and where RL methods typically fail because the exploration problem is too hard. We consider the Adroit tasks (Rajeswaran et al., 2017) represented in Figure 9, for which human demonstrations are available (25 episodes acquired using a virtual reality system). These environments come with a dense reward function that we replace with the following **sparse reward**: 1 if the goal is achieved, 0 otherwise.

**Algorithm & baselines.** The algorithm we propose is a two-fold training procedure: **1)** we learn a discretization of the action space in a fully offline fashion using the AQuaDem framework from human demonstrations; **2)** we train a discrete action deep RL algorithm on top of this this discretization. We refer to this algorithm as AQuaDQN. The RL algorithm considered is Munchausen DQN (Vieillard et al., 2020) as it is the state of the art on the Atari benchmark (Bellemare et al., 2013) (although we use the non-distributional version of it which simply amounts to DQN (Mnih et al., 2015) with a regularization term). To make as much use of the demonstrations as possible, we maintain two replay buffers: one containing interactions with the environment, the other containing the demonstrations that we sample using a fixed ratio similarly to DQfD (Hester et al., 2018), although we do not use the additional recipes of DQfD (multiple $n$-step evaluation of the bootstrapped estimate of $Q$, BC regularization term) for the sake of simplicity. When sampling demonstrations, the actions are discretized by taking the closest AQuaDem action candidate (using the Euclidean norm). We consider SAC and SAC from demonstrations (SACfD) –a modified version of SAC where demonstrations are added to the replay buffer (Vecerik et al., 2017)– as baselines against the proposed method. We do not include naive discretization baselines here, as the dimension of the action space is at least 24, which would lead to a $2^{24} \simeq 16M$ actions with a binary discretization scheme, which is prohibitive without additional assumptions on the structure of the action-value function.

**Evaluation & results.** We train the different methods on 1M environment interactions on 10 seeds for the chosen hyperparameters (a single set of hyperparameters for all tasks) and evaluate the agents every 50k environment interactions (without exploration noise) on 30 episodes. An episode is considered a success if the goal is achieved during the episode. The AQuaDem discretization is trained offline using 50k gradient steps on batches of size 256. The number of actions considered were $10, 15, 20$ and we found 10 to be performing best. Figure 4 shows the AQuaDem loss through the training procedure of the discretization step, and the Figure 5 shows the returns of the trained agents as well as their success rate. On Door, Pen, and Hammer, the AQuaDQN agent reaches high success rate, largely outperforming SACfD in terms of success and sample efficiency.

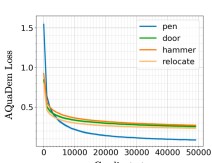

Figure 4: AQuaDem discretization loss.

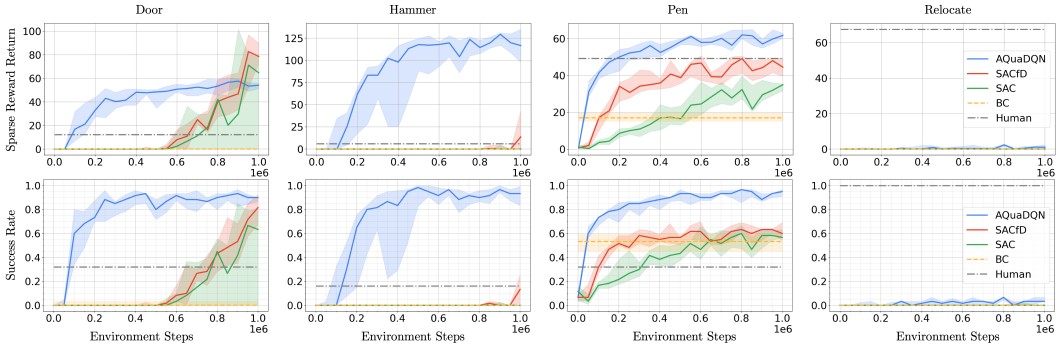

Figure 5: Performance of AQuaDQN against SAC and SACfD baselines. Agents are evaluated every 50k environment steps over 30 episodes. We represent the median performance in terms of success rate (bottom) and returns (top) as well as the interquartile range over 10 seeds.

On Relocate, all methods reach poor results (although AQuaDQN slightly outperforms the baselines). The task requires a larger degree of generalisation than the other three since the goal state and the initial ball position are changing at each episode. We show in Figure 6 that when tuned uniquely on the Relocate environment and with more environment interactions, AQuaDQN manages to reach a 50% success rate where other methods still fail. Notice that on the Door environment, the SAC and SACfD agents outperform the AQuaDQN agent in terms of final return (but not in term of success rate). The behavior of these agents are however different from the demonstrator since they consist in slapping the handle and abruptly pulling it back. We provide videos of all resulting agents (one episode for each seed which is not cherry picked) to demonstrate that AQuaDQN consistently learns a behavior that is qualitatively closer to the demonstrator.

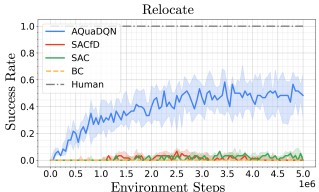

Figure 6: Performance of AQuaDQN against SAC and SACfD baselines when all are tuned on the Relocate environment. We represent the median performance in terms of success rate as well as the interquartile range over 10 seeds.

## 4.2 IMITATION LEARNING

**Setup.** In Imitation Learning, the task is not specified by the reward function but by the demonstrations themselves. The goal is to mimic the demonstrated behavior. There is no reward function and the notion of success is ill-defined (Hussenot et al., 2021). A number of existing works (Ho and Ermon, 2016; Ghasemipour et al., 2019; Dadashi et al., 2021) cast the problem into matching the state distributions of the agent and of the expert. Imitation Learning is of particular interest when designing a satisfying reward function –one that would lead the desired behavior to be the only optimal policy– is harder than directly demonstrating this behavior. In this setup, there is no reward provided, not in the environment interactions nor in the demonstrations. We again consider the Adroit environments and the human demonstrations which consist of 25 episodes acquired via a virtual reality system.

**Algorithm & baselines.** Again, the algorithm we propose has two stages. **1)** We learn –fully offline– a discretization of the action space using AQuaDem. **2)** We train a discrete action version of the GAIL algorithm (Ho and Ermon, 2016) in the discretized environment. More precisely, we interleave the training of a discriminator between demonstrations and agent experiences, and the training of a Munchausen DQN agent that maximizes the confusion of this discriminator. The Munchausen DQN takes one of the candidates actions given by AQuaDem. We call this algorithm AQuaGAIL. As a baseline, we consider the GAIL algorithm with a SAC (Haarnoja et al., 2018) agent directly maximizing the confusion of the discriminator. This results in a very similar algorithm as the one proposed by Kostrikov et al. (2019). We also include the results of BC (Pomerleau, 1991).

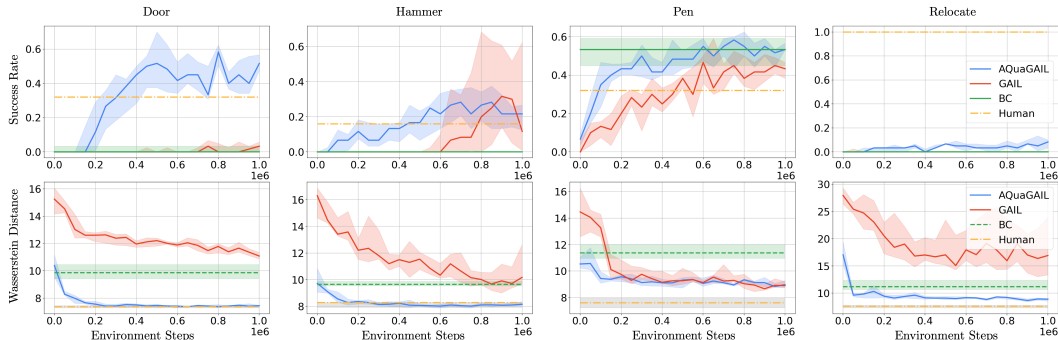

Figure 7: Performance of AQuaGAIL against GAIL and BC baselines. Agents are evaluated every 50k environment steps over 30 episodes. We represent the median success rate (top row) on the task as well as the Wasserstein distance (bottom row) of the agent's state distribution to the expert's state distribution as well as the interquartile range over 10 seeds.

**Evaluation & results.** We train AQuaGAIL and GAIL for 1M environment interactions on 10 seeds for the selected hyperparameters (a single set for all tasks). BC is trained for 60k gradient steps with batch size 256. We evaluate the agents every 50k environment steps during training (without exploration noise) on 30 episodes. The AQuaDem discretization is trained offline using 50k gradient steps on batches of size 256. The results are provided in Figure 7. Evaluating imitation learning algorithms has to be done carefully as the goal to "mimic a behavior" is ill-defined. Here, we provide the results according to two metrics. On top, the success rate is defined in Section 4.1. Notice that the human demonstrations do not have a success score of 1 on every task. We see that, except for Relocate, which is a hard task to solve with only 25 human demonstrations due to the necessity to generalize to new positions of the ball and the target, AQuaGAIL solves the tasks as successfully as the humans, outperforming GAIL and BC. Notice that our results corroborate previous work (Orsini et al., 2021) that showed poor performance of GAIL on human demonstrations after 1M steps. The second metric we provide, on the bottom, is the Wasserstein distance between the state distribution of the demonstrations and the one of the agent. We compute it using the POT library (Flamary et al., 2021) and use the Sinkhorn distance, a regularized version of the Wasserstein distance, as it is faster to compute. The "human" Wasserstein distance score is computed by randomly taking 5 episodes out of the 25 human demonstrations and compute the Wasserstein distance to the remaining 20. We repeat this procedure 100 times and plot the median (and the interquartile range) of the obtained values. Remark that AQuaGAIL is able to get much closer behavior to the human than BC and GAIL on all four environments in terms of Wasserstein distance. This supports that AQuaDem leads to policies much closer to the demonstrator. We provide videos of the trained agents as an additional qualitative empirical evidence to support this claim.

## 4.3 REINFORCEMENT LEARNING WITH PLAY DATA

**Setup.** The Reinforcement Learning with play data is an under-explored yet natural setup (Gupta et al., 2019). In this setup, the environment of interest has multiple tasks, a shared observation and action space for each task, and a reward function specific to each of the tasks. We also assume that we have access to *play data*, introduced by Lynch et al. (2020), which consists in episodes from a human demonstrator interacting with an environment with the sole intention to play with it. The goal is to learn an optimal policy for each of the tasks. We consider the Robodesk tasks (Kannan et al., 2021) shown in Figure 9, for which we acquired play data. We expand on the environment as well as the data collection procedure in the Appendix D.2.

**Algorithm & baselines.** Similarly to the RLfD setting, we propose a two-fold training procedure: **1)** we learn a discretization of the action space in a fully offline fashion using the AQuaDem framework on the play data, **2)** we train a discrete action deep RL algorithm using this discretization on each tasks. We refer to this algorithm as AQuaPlay. Unlike the RLfD setting, the demonstrations do not include any task specific reward nor goal labels meaning that we cannot incorporate the demonstration episodes in the replay buffer nor use some form of goal-conditioned BC. We use SAC as a baseline,

which is trained to optimize task specific rewards. Since the action space dimensionality is fairly low (5-dimensional), we can include naive uniform discretization baselines that we refer to as "bang-bang" (Bushaw, 1952). The original "bang-bang" controller (BB-2) is based on the extrema of the action space, we also provide a uniform discretization scheme based on 3 and 5 bins per action dimension, that we refer to as BB-3 and BB-5 respectively.

**Evaluation & results.**   We train the different methods on 1M environment interactions on 10 seeds for the chosen hyperparameters (a single set of hyperameters for all tasks) and evaluate the agents every 50k environment interactions (without exploration noise) on 30 episodes. The AQuaDem discretization is trained offline on play data using 50k gradient steps on batches of size 256. The number of actions considered were 10, 20, 30, 40 and we found 30 to be performing the best. It is interesting to notice that it is higher than for the previous setups. It aligns with the intuition that with play data, several behaviors needs to be modelled. The results are provided in Figure 8. The AQuaPlay agent consistently outperforms SAC in this setup. Interestingly, the performance of the BB agent decreases with the discretization granularity, well exemplifying the curse of dimensionality of the method. In fact, BB with a binary discretization (BB-2) is competitive with AQuaPlay, which validates that discrete action RL algorithms are well performing if the discrete actions are sufficient to solve the task. Note however that the Robodesk environment is a relatively low-dimensional action environment, making it possible to have BB as a baseline, which is not the case of *e.g.* Adroit where the action space is high-dimensional.

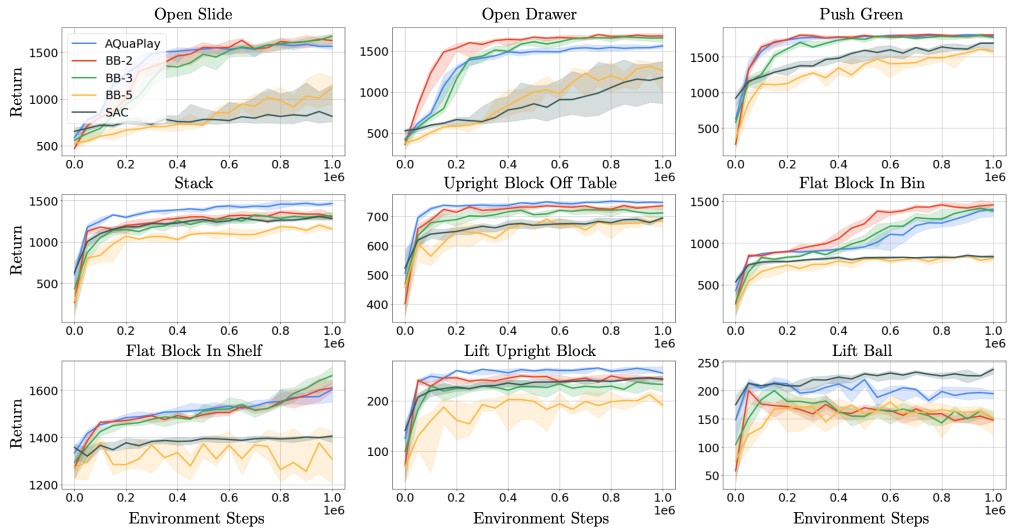

Figure 8: Performance of AQuaPlay against SAC and "bang-bang" baselines. Agents are evaluated every 50k environment steps over 30 episodes. We represent the median return as well as the interquartile range over 10 seeds.

## 5   RELATED WORK

**Continuous action discretization.**   The discretization of continuous action spaces has been introduced in control problems by Bushaw (1952) with the "bang-bang" controller (Bellman et al., 1956). This naive discretization is problematic in high-dimensional action spaces, as the number of actions grows exponentially with the action dimensionality. To mitigate this phenomenon, a possible strategy is to assume that action dimensions are independent (Tavakoli et al., 2018; Andrychowicz et al., 2020; Vieillard et al., 2021; Tang and Agrawal, 2020), or to assume or learn a causal dependence between them (Metz et al., 2017; Tessler et al., 2019; Sakryukin et al., 2020; Tavakoli et al., 2021). The AQuaDem framework circumvents the curse of dimensionality as the discretization is based on the demonstrations and hence is dependent on the multimodality of the actions picked by the demonstrator rather than the dimensionality of the action space. Close to our setup is the case where the action space is both discrete and continuous (Neunert et al., 2020) or the action space is discrete and large (Dulac-Arnold et al., 2015). Those setups are interesting directions for extending AQuaDem.

**Q-learning in continuous action spaces.** Policy-based methods consist in solving continuous or discrete MDPs based on maximizing the expected return over the parameters of a family of policies. If the return is estimated through Monte Carlo rollouts, this leads to algorithms that are typically sample-inefficient and difficult to train in high-dimensional action spaces (Williams, 1992; Schulman et al., 2015; 2017). As a result, a number of policy-based methods inspired from the policy gradient theorem (Sutton et al., 2000), aim at maximizing the return using an approximate version of the $Q$-value thus making them more sample-efficient. One common architecture is to parameterize a $Q$-value, which is estimated by enforcing Bellman consistency, and define a policy using an optimization procedure of the parametrized $Q$-value. Typical strategies to solve the $Q$-value maximization include enforcing the $Q$-value to be concave (Gu et al., 2016; Amos et al., 2017) making it easy to optimize through *e.g.* gradient ascent, to use a black box optimization method (Kalashnikov et al., 2018; Simmons-Edler et al., 2019; Lim et al., 2018), to solve a mixed integer programming problem (Ryu et al., 2020), or to follow a biased estimate of the policy gradient based on the approximate $Q$-value (Konda and Tsitsiklis, 2000; Lillicrap et al., 2016; Haarnoja et al., 2018; Fujimoto et al., 2018). Recently, Asadi et al. (2021) proposed to use a network that outputs actions together with their associated $Q$-values, tuned for each of the tasks at hand, on low-dimensional action spaces. Note that maximizing the approximate $Q$-value is a key problem that does not appear in discrete action space environments, thus justifying the interest of the AQuaDem framework.

**Hierarchical Imitation Learning.** A number of approaches have explored the learning of *primitives* or *options* from demonstrations together with a high-level controller that is either learned from demonstrations (Kroemer et al., 2015; Krishnan et al., 2017; Le et al., 2018; Ding et al., 2019; Lynch et al., 2020), or learned from interactions with the environment (Manschitz et al., 2015; Kipf et al., 2019; Shankar et al., 2019), or hand specified (Pastor et al., 2009; Fox et al., 2019). AQuaDem can be loosely interpreted as a two-level procedure as well, where the primitives (action discretization step) are learned fully offline, however there is no concept of goal nor temporally extended actions.

**Modeling multimodal demonstrations.** A number of works have modeled the demonstrator data using multimodal architectures. For example, Chernova and Veloso (2007); Calinon and Billard (2007) introduce Gaussian mixture models in their modeling of the demonstrator data. More recently, Rahmatizadeh et al. (2018) use Mixture density networks together with a recurrent neural network to model the temporal correlation of actions as well as their multimodality. (Yu et al., 2018) also uses Mixture density networks to meta-learn a policy from demonstrations for one-shot adaptation. Another recent line of works has considered the problem of modeling demonstrations using an energy-based model, which is well adapted for multimodalities (Jarrett et al., 2020; Florence et al., 2021). Singh et al. (2020) also exploit the demonstrations prior for downstream tasks by learning a prior using a state-conditioned action generative model coupled with a continuous action algorithm. This is different from AQuaDem that exploits the demonstrations prior to learn a discrete action space in order to use discrete action RL algorithms.

## 6 PERSPECTIVES AND CONCLUSION

With the AQuaDem paradigm, we provide a simple yet powerful method that enables to use discrete-action deep RL methods on continuous control tasks using demonstrations, thus escaping the complexity or curse of dimensionality of existing discretization methods. We showed in three different setups that it provides substantial gains in sample efficiency and performance and that it leads to qualitatively better agents, as enlightened by the videos provided in the supplementary material.

There are a number of different research avenues opened by AQuaDem. Other discrete action specific methods could be leveraged in a similar way in the context of continuous control: count-based exploration (Tang et al., 2017), planning (Browne et al., 2012) or offline RL (Lagoudakis and Parr, 2003; Riedmiller, 2005). Similarly a number of methods in Imitation Learning (Brantley et al., 2019; Wang et al., 2019) or in offline RL (Fujimoto and Gu, 2021; Wu et al., 2019) are evaluated on continuous control tasks and are based on Behavioral Cloning regularization which could be refined using the same type of multioutput architecture used in this work. Another possible direction for the AQuaDem framework is to be analyzed in the light of risk-MDPs as the constraint of the action space arguably reduces a notion of risk when acting in this environment. Finally, as the gain of sample efficiency is clear in different experimental settings, we believe that the AQuaDem framework could be an interesting avenue for learning controllers on physical systems.

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

## A    CONNECTION TO GAUSSIAN MIXTURE MODELS

The BC loss can be interpreted as a maximum likelihood objective under the assumption that the demonstrator data comes from a Gaussian distribution. Similarly to Mixture density networks (Bishop, 1994), we propose to replace the Gaussian distribution by a mixture of Gaussian distributions. Suppose we represent the probability density of an action conditioned on a state by a mixture of $K$ Gaussian kernels: $p(a|s) = \sum_{k=1}^{K} \alpha_k(s) d_k(a|s)$, where $\alpha_k(s)$ is the mixing coefficient (that can be interpreted as a state conditioned prior probability), and $d_k(a|s)$ is the conditional density of the target $a$. Now assuming that the $K$ kernels are centered on $\Psi_k(s)_{k=1:K}$ and have fixed covariance $\sigma^2 \mathbb{1}$ where $\sigma$ is a hyperparameter, we can write the log-likelihood of the demonstrations data $\mathcal{D}$ as:

$$\mathcal{LL}(\mathcal{D}) = \sum_{s,a \in \mathcal{D}} \log(p(s)p(a|s)) = \sum_{s,a \in \mathcal{D}} \log p(s) + \log\big(\sum_{k=1}^{K} \alpha_k(s) d_k(a|s)\big)$$

$$= \sum_{s,a \in \mathcal{D}} \log p(s) + \log\big(C \sum_{k=1}^{K} \alpha_k(s) \exp\big(-\frac{\|\Psi_k(s) - a\|^2}{\sigma^2}\big)\big).$$

Therefore minimizing the negative log likelihood reduces to minimizing:

$$\sum_{s,a \in \mathcal{D}} -\log\big(\sum_{k=1}^{K} \alpha_k(s) \exp\big(-\frac{\|\Psi_k(s) - a\|^2}{\sigma^2}\big)\big).$$

We propose to use a uniform prior $\alpha_k(s) = \frac{1}{K}$ when learning the locations of the centroids, which leads exactly to Equation (1) where the variance $\sigma^2$ is the temperature $T$. Note that we initially learned the state conditioned prior $\alpha_k(s)$, but we found no empirical evidence that it may be used to improve the performance of the downstream algorithms defined in Section 4.

## B    ACTION VISUALIZATION IN A HIGH-DIMENSIONAL ENVIRONMENT

For the Door environment (see Figure 9) we represent the actions candidates learned using the AQuaDem framework with videos that can be found in the supplementary material in the folder `visualizations` (for the best hyperparameters in the RL with demonstrations setting see Appendix D.4.1). As the action space is of high dimensionality, we choose to represent each action dimension on the x-axis, and the value for each dimension on the y-axis. We connect the dots on the x-axis to facilitate the visualization through time. We replay a trajectory from the human demonstrations and show at each step the 10 actions proposed by the AQuaDem network, and the action actually taken by the human demonstrator. Each action candidate has a color consistent across time (meaning that the blue action always correspond to the same head of the $\Psi$ network). Interestingly, the video shows that actions are very state dependent (except some default 0-action) and evolve smoothly through time.

## C    ABLATION STUDY

In this section, we provide two ablations of the AQuaDQN algorithm. The first ablation is to learn a fixed set of actions independently of the state (which reduces to $K$-means). The second ablation consists in using random actions rather than the actions learned by the AQuaDem framework (the actions are given by the AQuaDem network, randomly initialized and not trained). We use the same hyperparameters as the one selected for AQuaDQN. In each case, for a number of actions in $\{5, 10, 25\}$, the success rate of the agent is 0 for all tasks throughout the training procedure.

## D    IMPLEMENTATION

### D.1    GRID WORLD VISUALIZATIONS

We learn the discretization of the action space using the AQuaDem framework. The architecture of the network is a common hidden layer of size 256 with relu activation, and a subsequent hidden layer

of size 256 with relu activation for each action. We minimize the AQuaDem loss using the Adam optimizer with the learning rate 0.0003 and the dropout regularization rate 0.1 on 20000 gradient steps.

## D.2 ENVIRONMENTS

We considered the Adroit environments and the Robodesk environments, for which we described the observation space and the action space in Table 1.

| Environment | Observation Space | Action Space |
|---|---|---|
| Door | 39 | 28 |
| Hammer | 46 | 26 |
| Pen | 45 | 24 |
| Relocate | 39 | 30 |
| Robodesk | 76 | 5 |

Table 1: Environment description of the Adroit and Robodesk observation and action space.

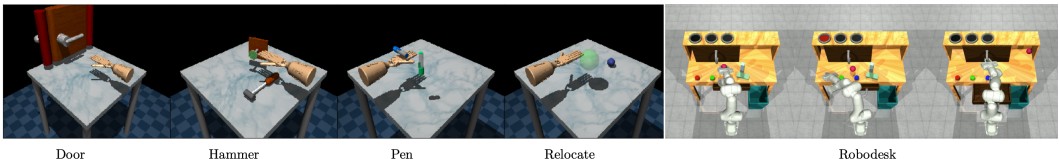

Door        Hammer        Pen        Relocate        Robodesk

Figure 9: Visualizations of the Adroit and Robodesk environments.

**Adroit**   The Adroit environments (Rajeswaran et al., 2017) consists in a shadow hand solving 4 tasks (Figure 9). The environments come with demonstrations which are gathered using virtual reality by a human.

**Robodesk**   The Robodesk environment (Kannan et al., 2021) consists of a simulated Franka Emika Panda robot interacting with a table where multiple tasks are possible. The version of the simulated robot in the Robodesk environment only includes 5 DoFs (vs the 7 DoFs available, 2 were made not controllable). We evaluate AQuaPlay on the 9 base tasks described in Robodesk: `open_slide`, `open_drawer`, `push_green`, `stack`, `upright_block_off_table`, `flat_block_in_bin`, `flat_block_in_shelf`, `lift_upright_block`, `lift_ball`.

We used the RLDS creator `github.com/google-research/rlds-creator` to generate play data, together with a Nintendo Switch Pro Controller. The data is composed by 50 episodes of approximately 3 minutes where the goal of the demonstrator is to *interact* with the different elements of the environment.

## D.3 HYPERPARAMETER SELECTION PROCEDURE

In the section we provide the hyperparameter selection procedure for the different setups. For the RLfD setting (Section 4.1) and the IL setting (Section 4.2) the number of hyperparameters is prohibitive to perform grid search. Therefore, we propose to sample hyperparameters uniformly within the set of all possible hyperparameters. For each environment, we sample 1000 configurations of hyperparameters, and train each algorithm including the baselines. We compute the average success rate of each individual value on the top 50% of all corresponding configurations (since poorly performing configurations are less informative) and select the best performing hyperparameter value independently. This procedure enables to **1)** limit combinatorial explosion with the number of hyperparameters **2)** provide a fair evaluation between the baselines and the proposed algorithms as they all rely on the same amount of compute. In the supplementary material, we provide histograms detailing the influence of each hyperparameter. For the RLfP setting, we fixed the parameters related to the DQN algorithm with the ones selected in the RLfD setting to limit the hyperparameter search, which enables to perform grid search for 3 seeds, and select the best set of hyperparameters.

### D.4 REINFORCEMENT LEARNING WITH DEMONSTRATIONS

#### D.4.1 AQUADQN

We learn the discretization of the action space using the AQuaDem framework. The architecture of the network is a common hidden layer of size 256 with relu activation, and a subsequent hidden layer of size 256 with relu activation for each action. We minimize the AQuaDem loss using the Adam optimizer and dropout regularization.

We train a DQN agent on top of the discretization learned by the AQuaDem framework. The architecture of the Q-network we use is the default LayerNorm architecture from the Q-network of the ACME library (Hoffman et al., 2020), which consists in a hidden layer of size 512 with layer normalization and tanh activation, followed by two hidden layers of sizes 512 and 256 with elu activation. We explored multiple $Q$-value losses for which we used the Adam optimizer: regular DQN (Mnih et al., 2015), double DQN with experience replay (Van Hasselt et al., 2016; Schaul et al., 2016), and Munchausen DQN (Vieillard et al., 2020); the latter led to the best performance. We maintain a fixed ratio of demonstration episodes and agent episodes in the replay buffer similarly to Hester et al. (2018). We also provide as a hyperparameter an optional minimum reward to the transitions of the expert to have a denser reward signal. The hyperparameter sweep for AQuaDQN can be found in Table 2. The complete breakdown of the influence of each hyperparameter is provided in `hps_lfd_aquadqn.html` in the supplementary.

| Hyperparameter | Possible values |
|---|---|
| aquadem learning rate | 3e-5, 0.0001, **0.0003**, 0.001, 0.003 |
| aquadem input dropout rate | 0, **0.1**, 0.3 |
| aquadem hidden dropout rate | 0, **0.1**, 0.3 |
| aquadem temperature | 0.0001, **0.001**, 0.01 |
| aquadem # actions | **10**, 15, 20 |
| dqn learning rate | 0.00003, **0.0001**, 0.003 |
| dqn n step | 1, **3**, 5 |
| dqn epsilon | 0.001, 0.01, **0.1** |
| dqn ratio of demonstrations | 0, 0.1, **0.25**, 0.5 |
| dqn min reward of demonstrations | None, **0.01** |

Table 2: Hyperparameter sweep for the AQuaDQN agent

When selecting hyperparameters specifically for Relocate, for Figure 6, the main difference in the chosen values is a dropout rate set to 0.

#### D.4.2 SAC AND SACFD

We reproduced the authors' implementations (with an adaptive temperature) and use MLP networks for both the actor and the critic with two hidden layers of size 256 with relu activation. We use an Adam optimizer to train the SAC losses. We use a replay buffer of size 1M, and sample batches of size 256. We introduce a parameter of gradient updates frequency $n$ which indicates a number of $n$ gradient updates on the SAC losses every $n$ environment steps. SACfD is a version of SAC inspired by DDPGfD (Vecerik et al., 2017) where we add expert demonstrations to the replay buffer of the SAC agent with a ratio between the agent episodes and the demonstration episodes which is a hyperparameter. We also provide as a hyperparameter an optional minimum reward to the transitions of the expert to have a denser reward signal. We found that the best hyperparameters for SAC are the same for SACfD. The HP sweep for SAC and SACfD can be found in Table 3 and Table 4. The complete breakdown of the influence of each hyperparameter is provided in `hps_lfd_sac.html` and `hps_lfd_sacfd.html` in the supplementary.

| Hyperparameter | Possible values |
|---|---|
| learning rate | 3e-5, **1e-4**, 0.0003 |
| n step | 1, 3, **5** |
| tau | **0.005**, 0.01, 0.05 |
| reward scale | 0.1, 0.3, **0.5** |

Table 3: Hyperparameter sweep for SAC.

| Hyperparameter | Possible values |
|---|---|
| learning rate | 3e-5, **1e-4**, 0.0003 |
| n step | 1, 3, **5** |
| tau | **0.005**, 0.01, 0.05 |
| reward scale | 0.1, **0.3**, 0.5 |
| ratio of demonstrations | 0, **0.001**, 0.1, 0.25 |
| mini reward of demonstrations | None, **0.01**, **0.1** |

Table 4: Hyperparameter sweep for SACfD.

## D.5 IMITATION LEARNING

### D.5.1 AQUAGAIL

We learn the discretization of the action space using the AQuaDem framework. The architecture of the network is a common hidden layer of size 256 with relu activation, and a subsequent hidden layer of size 256 with relu activation for each action. We minimize the AQuaDem loss using the Adam optimizer and dropout regularization. The discriminator is a MLP whose number of layers, number of units per layers are hyperparameters. We use the Adam optimizer with two possible regularization scheme: dropout and weight decay. The discriminator outputs a value $p$ from which we compute three possible rewards $-\log(p), -0.5\log(p) + \log(1-p), \log(1-p)$ corresponding to the reward balance hyperparameter. The direct RL algorithm is Munchausen DQN, with the same architecture and hyperparameters described in Section D.4.1. The hyperparameter sweep for AQuaGAIL can be found in Table 5. The complete breakdown of the influence of each hyperparameter is provided in `hps_il_aquagail.html` in the supplementary.

| Hyperparameter | Possible values |
|---|---|
| discriminator learning rate | 1e-7, 3e-7, **1e-6**, 3e-5, 1e-4 |
| discriminator num layers | **1**, 2 |
| discriminator num units | 16, **64**, 256 |
| discriminator regularization | none, **dropout**, weight decay |
| discriminator weight decay | 5, 10, 20 |
| discriminator input dropout rate | **0.5**, 0.75 |
| discriminator hidden dropout rate | **0.5**, 0.75 |
| discriminator observation normalization | **True**, False |
| discriminator reward balance | 0., **0.5**, 1. |
| dqn learning rate | **3e-5**, 1e-4, 3e-4 |
| dqn n step | **1**, 3, 5 |
| dqn epsilon | 0.001, **0.01**, 0.1 |
| aquadem learning rate | 3e-5, 1e-4, **3e-4**, 1e-3, 3e-3 |
| aquadem temperature | 0.0001, **0.001**, 0.01 |
| aquadem num actions | **10**, 15, 20 |
| aquadem input dropout rate | **0**, 0.1, 0.3 |
| aquadem hidden dropout rate | **0**, 0.1, 0.3 |

Table 5: Hyperparameter sweep for the AQuaGAIL agent.

### D.5.2 GAIL

We used the same discriminator architecture and hyperparameters as the one described in Section D.5.1. The direct RL agent is the SAC algorithm whose architecture and hyperparameters are described in Section D.4.2. The hyperparameter sweep for GAIL can be found in Table 6. The complete breakdown of the influence of each hyperparameter is provided in `hps_il_gail.html` in the supplementary.

| Hyperparameter | Possible values |
|---|---|
| discriminator learning rate | 1e-7, **3e-7**, 1e-6, 3e-5, 1e-4 |
| discriminator num layers | **1**, 2 |
| discriminator num units | 16, 64, **256** |
| discriminator regularization | none, dropout, **weight decay** |
| discriminator weight decay | 5, **10**, 20 |
| discriminator input dropout rate | 0.5, 0.75 |
| discriminator hidden dropout rate | 0.5, 0.75 |
| discriminator observation normalization | **True**, False |
| discriminator reward balance | 0., **0.5**, 1. |
| sac learning rate | 3e-5, **1e-4**, 3e-4 |
| sac n step | 1, 3, **5** |
| sac tau | 0.005, 0.01, **0.05** |
| sac reward scale | 0.1, 0.3, **0.5** |

Table 6: Hyperparameter sweep for the discriminator part of the GAIL agent.

### D.5.3 BEHAVIORAL CLONING

The BC network is a MLP whose number of layers, number of units per layers and activation functions are hyperparameters. We use the Adam optimizer with two possible regularization scheme: dropout and weight decay. The observation normalization hyperparameter is set to True when each dimension of the observation are centered with the mean and standard deviation of the observations in the demonstration dataset. The complete breakdown of the influence of each hyperparameter is provided in `hps_il_bc.html` in the supplementary.

| Hyperparameter | Possible values |
|---|---|
| learning rate | 1e-5, 3e-5, 1e-4, **3e-4**, 1e-3 |
| num layers | **1**, 2. 3 |
| num units | 16, 64, **256** |
| activation | relu, **tanh** |
| observation normalization | **True**, False |
| weight decay | **0**, .01, 0.1 |
| input dropout rate | **0**, 0.15, 0.3 |
| hidden dropout rate | **0**, 0.25, 0.5 |

Table 7: Hyperparameter sweep for the BC agent.

### D.6 REINFORCEMENT LEARNING WITH PLAY DATA

### D.6.1 SAC

We used the exact same implementation as the one described in Section D.4.2. The HP sweep can be found in Table 8.

### D.6.2 DQN WITH NAIVE DISCRETIZATION

We used the exact same implementation as the one described in Section D.4.1, and also use the best hyperparameters found in the RLfD setting. As the action space is $(-1, 1)^5$, we use three different

| Hyperparameter | Possible values |
|---|---|
| learning rate | 1e-5, 3e-5, 3e-4, **1e-4** |
| n step | 1, 3, **5** |
| reward scale | 0.1, **1**, 10 |
| tau | 0.005, **0.01**, 0.05 |

Table 8: Hyperparameter sweep for SAC for the Robodesk environment. The best hyperparameter set was chosen as the one that maximizes the performance on average on all tasks.

discretization meshes: $\{-1, 1\}, \{-1, 0, 1\}, \{-1, -0.5, 0., 0.5, 1\}$ which induce a discrete action space of dimension $2^5, 3^5, 5^5$ respectively. We refer to the resulting algorithm as BB-2, BB-3, and BB-5 (where BB stands for "Bang-bang").

### D.6.3 AQUAPLAY

We used the exact same implementation as the one described in Section D.4.1, and also use the best hyperparameters found in the RLfD setting for the Munchausen DQN agent. We performed a sweep on the discretization step that we report in Table 9.

| Hyperparameter | Possible values |
|---|---|
| learning rate | 0.0001, 0.0003, **0.001** |
| dropout rate | 0, **0.1**, 0.3 |
| temperature | **1e-4**, 1e-3, 1e-2 |
| # actions | 10, 20, **30**, 40 |

Table 9: Hyperparameter sweep for the AQuaPlay agent. The best hyperparameter set was chosen as the one that maximizes the performance on average on all tasks.

## E  SANITY CHECK BASELINES

We provide in Figure 10 the results of our SAC implementation on the 5 classical OpenAI Gym environments, for 20M steps. The hyperparameters are the one of the original papers (original paper, followed by the introduction of the adaptative temperature). The results correspond to what is provided in the literature.

We provide in Figure 11 the results on OpenAI Gym environment sof the configurations of SAC and AQuaDQN that were selected for Adroit (and not Mujoco!). The action discretization is learned on the medium-expert dataset from D4RL (no dataset is provided for Humanoid and that is why we do not include it in this figure). One can see in Figure 10 that SAC can perform much better on Mujoco Gym after 1M steps when selecting hyperparameters on the environments themselves.

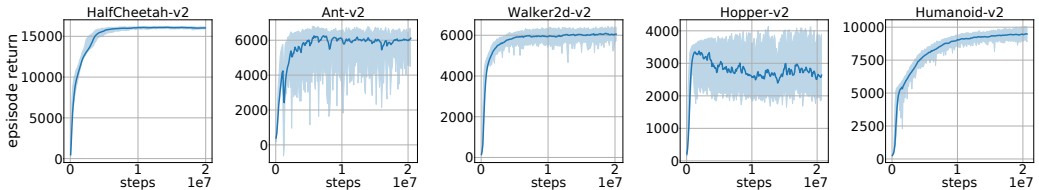

Figure 10: SAC median and interquartile range on 10 seeds on the 5 Open Gym environments.

## F  PROOF

For lighter notations, we write $x_k = ||\Psi_k(s) - a||^2$ and $x = (x_1, ..., x_K)$. For a single state-action pair (the empirical expectation being not relevant for studying the effect of the temperature), the

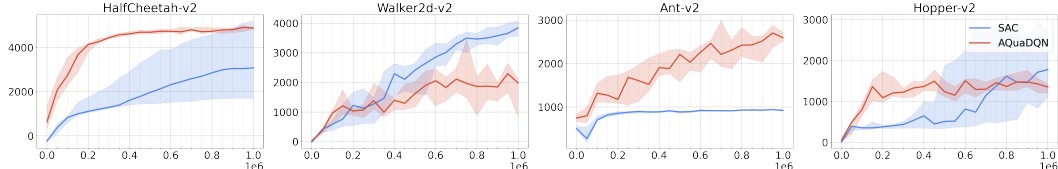

Figure 11: SAC and AQuaDQN medians and interquartile ranges on 10 seeds on the Open Gym environments. These configurations are the ones that were selected to perform best on Adroit and not for Mujoco. One can see in Figure 10 that SAC can perform much better on Mujoco Gym after 1M steps when selecting hyperparameters on the environments themselves.

AQuaDem loss can be rewritten:

$$J(\Psi) = -\log \sum_{k=1}^{K} \exp(-\frac{x_k}{T})$$

$$= \frac{1}{T}(-T \log \sum_{k=1}^{K} \exp(-\frac{x_k}{T}))$$

$$= \frac{1}{T}(-T \log(\frac{1}{K} \sum_{k=1}^{K} \exp(-\frac{x_k}{T}))) - \log K.$$

Let's define $f_T(x) = -T \log(\frac{1}{K} \sum_{k=1}^{K} \exp(-\frac{x_k}{T}))$. The function $f_T$ is the same as the loss up to (1) a constant term that does not change the solution of the optimization problem, (2) a scaling term that can be absorbed into the learning rate. So, we can study this function for the behavior of the loss with the respect to the temperature.

Now, denoting $x_m = \min_k x_k$, we'll first study the behavior for low temperature.

$$f_T(x) = T \log K - T \log(\sum_{k=1}^{K} \exp(-\frac{x_k}{T}))$$

$$= T \log K - T \log(\exp(\frac{-x_m}{T}) \sum_{k=1}^{K} \exp(-\frac{x_k - x_m}{T}))$$

$$= T \log K + x_m - T \log(1 + \sum_{k=1,k\neq m}^{K} \exp(-\frac{x_k - x_m}{T}))$$

$$\xrightarrow[T\to 0]{} x_m.$$

Therefore, when the temperature goes to zero, $f_T$ behaves as the minimum.

For large temperatures, we have, using Taylor expansions:

$$f_T(x) = -T \log(\sum_{k=1}^{K} \frac{1}{K} \exp(-\frac{x_k}{T}))$$

$$= -T \log(\sum_{k=1}^{K} \frac{1}{K}(1 - \frac{x_k}{T} + o(\frac{x_k}{T})))$$

$$= -T \log(1 - \frac{1}{K} \sum_{k=1}^{K} \frac{x_k}{T} + o(\frac{1}{T}))$$

$$= \frac{T}{K} \sum_{k=1}^{K} \frac{x_k}{T} + O(\frac{1}{T})$$

$$\xrightarrow[T \to \infty]{} \frac{1}{K} \sum_{k=1}^{K} x_k.$$

So, when the temperature goes to infinite, $f_T$ behaves as the average.

