# OpenReview forum: "Continuous Control with Action Quantization from Demonstrations"
_ICLR.cc/2022/Conference — ICLR 2022 Submitted_

### Official Review · Reviewer_hp36 · 2021-10-18

**Correctness:** 3
**Technical Novelty And Significance:** 2
**Empirical Novelty And Significance:** 2
**Recommendation:** 5
**Confidence:** 5

**Main Review:**

* This paper proposed an action space discretization method, it claims to not suffer from the curse of dimensionality and not needing any specific assumptions about the task. I agree with the author that this is an important topic that RL community should keep an eye on. However, I might not agree that discrete control problems are harder than continuous control problems. I think the argument in the introduction that since continuous control problem is much harder than discrete ones, so we should turn a continuous problem into a discrete problem is debatable.

* The main idea is to capture the mode of continuous actiosn taken under different states through a Gaussian Mixture Model type of neural network. The key novelty is Equation 1. where Figure 2 explains the key idea very well.

* On the empirical side, the author has considered three settings: RL with demonstration, RL with play data, and imitation learning. The significance of the results is strong.

**Summary Of The Paper:**

This paper proposes a novel method: Action Quantization from Demonstrations (AQuaDem) to learn a discretization of continuous action spaces by leveraging the priors of demonstrations. As such, one can apply any discrete action deep RL algorithm to continuous problems. In three scenarios: RL with demonstration, RL with play data and imitation learning, the effectiveness is validated.

**Summary Of The Review:**

 I think the technical side of the story is sound, the method taken is reasonable and the empirical results are valid. However, I think the novelty of this paper is limited. The idea is rather simple, and learning discretization through a gaussian mixture model type of representation is trivial. I am not quite convinced at the current stage that it reaches an ICLR level of work yet.

---

> ### Author Response · Authors · 2021-11-10
> **Answer 1/1**
>
> Thanks for the review.
>
> **“ [...] so we should turn a continuous problem into a discrete problem is debatable.”**
>
> We would be happy to enter into that debate with the reviewer and maybe be less affirmative in our intro about this specific claim. But as the reviewer states, this is anyway an important topic that the RL community should keep an eye on.
> We hopefully contributed to that debate  by providing some empirical evidence in the paper (like the bang bang controller experiments –without additional data–, or all experiments using AQuaDem–with additional data. We would be thankful to the reviewer to receive counter arguments that we could use to strengthen our intro.
>
> **"The idea is rather simple, and learning discretization through a gaussian mixture model type of representation is trivial."**
>
> Actually, we consider simplicity as an asset for the method. We strongly disagree on the lack of novelty but we would certainly consider any supporting evidence of this opinion (eg, references).
>
> **I am not quite convinced at the current stage that it reaches an ICLR level of work yet.**
>
> We would gladly welcome concrete and constructive feedback to support this statement. This method has to the best of our knowledge not been tried before, it works very well and makes it possible to solve some tasks never solved before. On top of that, it is simple which is definitely an advantage. We think we shouldn't judge the complexity of the solution but rather the complexity of the problem we tackle. We argue we provide a simple solution to a rather complex problem.

---

> ### Author Response · Authors · 2021-11-17
> **Message to reviewer**
>
> Dear reviewer, there're only 5 days left before the end of discussion, and no interaction so far. We would be more than happy to know if our rebuttal and revision addressed your questions and comments, or, if not, what are the remaining concerns you may have.
>
> Best,
>
> The authors

---

> > ### Comment · Reviewer_hp36 · 2021-12-01
> > **I maintain my score of evaluation.**
> >
> > The main evaluation is that I think the novelty is limited, in the sense that learning discretization through a gaussian mixture model type of representation is trivial

---

> > > ### Author Response · Authors · 2021-12-01
> > > **Answer to the reviewer**
> > >
> > > Thank you for your response.
> > >
> > > We encourage the reviewer to realize that 1) the simplicity of a method is desirable 2) novelty and simplicity are two orthogonal principles 3) the novelty in our paper is not based on the architecture of the method (which is simple, which again is a good thing) but on the use of demonstrations to quantize an action space (see the review of Reviewer MjVt).

---

### Official Review · Reviewer_MjVt · 2021-10-31

**Correctness:** 3
**Technical Novelty And Significance:** 3
**Empirical Novelty And Significance:** 4
**Recommendation:** 6
**Confidence:** 5

**Main Review:**

## Strengths:
1) Repurposing BC and extending it to serve as a discrete action-space learner is interesting.
2) Paper is generally well written, and easy to follow.
3) Choices of methods to combine the proposed approach with and the environments to test in are reasonable.
4) The breadth of discussing the related work is reasonable.

---
## Weaknesses:
1) Some important missing baselines (some of which are referred to briefly but somewhat dismissed in experiments).
2) Some missing related works.
3) Presentation at some points is misleading or incomplete.
4) Some discussions missing.

---
## Issues:
- **Sec. 1, paragraph 2** "*[in finite discrete action spaces] exploration is more manageable [and] computing the maximum of the action-value function is straightforward*"**:** A finite discrete action space can have a much higher intrinsic complexity than a continuous action space. As such, I wouldn't say that in general exploration is more manageable. For instance, finite but large combinatorial action spaces can be much harder to explore with action-composite methods (such as standard DQN) as opposed to when a continuous embedding can be assumed and exploited using PG methods (see, e.g., Dulac-Arnold et al. (2015)). In addition, the exponential complexity of maximization in combinatorial discrete action spaces can fast become computationally intractable --- it isn't straightforward in general.

- **Sec. 1, paragraph 2:** The end of this paragraph seems to imply two algorithmic choices for continuous-action control: (1) MC/Reinforce-style, (2) Q-learning-style. This could be written much better in my view to encapsulate the full landscape of approaches better (e.g. actor-critic methods). Otherwise, clarify what is meant.

- **Bang-bang controller:** BB controller implies a very specific type of discretization scheme, where only the extrema of the action space are used in the control problem. However, in this paper, you use this to refer to the generic uniform discretization scheme. Not sure if referring to this scheme as BB is really correct or helpful.

- **Sec. 1, paragraph 3:** Andrychowicz et al. (2020) does not assume a causal dependence between action dimensions as far as I know. If they do, please explain in your response. Also, I believe this paragraph is a good place to also refer to approaches that assume independence between action dimensions: (Tavakoli et al., 2018; Tang and Agrawal, 2020; Andrychowicz et al., 2020).

- **Sec. 1, paragraph 3** "*[...] but they are typically complex and task-specific.*" **:** Most of these approaches are trivial to implement. Regarding task-specificity, they are no more task-specific than standard model-free methods such as DQN and DDPG are. Regarding being complex, could you clarify in what respect it is meant? Regarding task-specificity, I personally understand that your method relies less on the task specification, but do state more clearly for the reader in what way your method is less task-specific.

- **Sec. 2, paragraphs 1-2:** *r* is used to denote both a *random variable* (when discussing the Q-learning update rule with sampled transitions) and the *reward function* (e.g. when expressing the recursive form of $Q^\pi$ and $V^\pi$). Also, while often in papers it is omitted that by "reward function" in fact the *expected* reward function is meant, it is useful to state that explicitly.

- **Sec. 4.1, paragraph 2 (last line):** Action dimensionality of 24 is manageable by all following approaches:
  - Factored DQN methods (Metz et al., 2017, Tavakoli et al., 2018) and Refs. [2, 3].
  - Factored PG methods (Tang and Agrawal, 2020).

- **Sec. 4.3, Evaluation & results** "*Interestingly, the performance of the BB agent decreases with the discretization granularity, well exemplifying the curse of dimensionality of the method.*" **:** This has been discussed in Ref. [2], where exploiting compositionality has also been shown to be key in solving this problem. At the end of the same paragraph, again it is mentioned that the reason why uniform discretization is not used is that it is not possible to run them in high-dim. action spaces. These statements seem to ignore the line of works discussed at the beginning of your paper's Related Work. Is there a reason for this?

- **Sec. 5, paragraph 1:**
  - Insert (Tang and Agrawal, 2020): "*[...] action dimensions are independent (Tavakoli et al., 2018; **Tang and Agrawal, 2020**, Andrychowicz et al., 2020 [...])*".
  - Remove (Tang and Agrawal, 2020) from *autoregressive discretization* citations.
  - After referencing methods that assume independence or those assuming some specific causal dependence, refer also to methods that learn the causal dependence (Ref. [2,4]).

---
## Questions:
1. Please respond to the questions posed in the *Issues* section of my review.

2. Why not compare with action representation learning methods (see, e.g., Ref. [1])? This could be quite a valuable comparison, and several possibilities could be explored.

3. Why only try the discrete-action baseline in the low-dimensional case of the Robodesk domain? As you have noted in the related work, there are now a few approaches for scaling discrete-action Q-based and PG methods to high dimensional problems; why not try them?

4. **Figure 6:** How does the tuning process on Relocate work?

5. What is the difference between *Success Rate* and *Sparse Reward Returns* in Fig. 5? Given that sparse rewards are 1 if the goal is achieved, I expected these two to be the same.

6. **Table 8 (and experiments in general):** Choosing hyperparameters that perform best on average across all tasks (including test ones) is not fair. Am I misunderstanding something here, or is there a reason why this makes sense here?

7. Could you position your work w.r.t. the line of work of action-embedding/action-representation/latent-action learners (e.g. in the context of Refs. [1,5])?

---
### References:

[1] Chandak, Y., Theocharous, G., Kostas, J., Jordan, S., Thomas, P. (2019). Learning Action Representations for Reinforcement Learning. In *Proceedings of the 36th International Conference on Machine Learning*.

[2] Tavakoli, A., Fatemi, M., Kormushev, P. (2021). Learning to Represent Action Values as a Hypergraph on the Action Vertices. In *Proceedings of the Ninth International Conference on Learning Representations*.

[3] Van de Wiele, T., et al. (2020). Q-learning in enormous action spaces via amortized approximate maximization. *arXiv preprint arXiv:2001.08116*.

[4] Sakryukin, A., Raissi, C., Kankanhalli, M. (2020). Inferring DQN structure for high-dimensional continuous control. In *Proceedings of the 37th International Conference on Machine Learning*.

[5] Merel, J., et al. (2019). Neural probabilistic motor primitives for humanoid control. In *Proceedings of the Seventh International Conference on Learning Representations*.




**Summary Of The Paper:**

This paper proposes a method for discrete action-space learning from originally continuous action spaces using demonstrations. In other words, the aim is to utilize demonstration data so as to find a set of K discrete actions in a given state. If expert demonstrations represent an individual greedy optimal policy, the demonstrations sufficiently cover the state space, and the demo-generating optimal policy and the learning agent's policy are fully observing of the state (no partial observability), then K=1 can be sufficient to yield the greedy demo-generating optimal policy. However, in reality, these assumptions are hardly ever fulfilled. In addition, the demonstration data may not even come from an *optimal policy* for a task (e.g. human demonstration is generally suboptimal), or be w.r.t. to any specific downstream task in mind at all (*free play*). In such general cases, learning a discrete subspace of the continuous action space is quite useful. First, it allows us to apply discrete-action methods instead of continuous-action ones (which have been recently shown to be better even without incorporating demonstration data). Second, the reduced action space makes exploration significantly easier. Regarding the latter, this method can be seen as learning a hard bias on exploration (removing a subset of actions from the original downstream action space) and only using those used more frequently in demonstrations. The method used to achieve this is similar in spirit to Behavioral Cloning (BC) and, in fact, becomes equivalent to BC in the extreme case of K=1.

**Summary Of The Review:**

I generally like this paper: the proposed method is sound and useful, experiments are insightful and nicely designed, and good discussions around limitations (e.g. pruning optimal actions) are present.

However, the paper makes assertions here and there that are not entirely true (e.g. while related work covers a group of methods for addressing the curse of dimensionality, on various occasions this line of work is dismissed and a limited set of baselines are used). Baselines that use uniform discretization and can scale to high dimensions (regarding computation and generalization capabilities) by exploiting compositionality in multi-dimensional spaces are simple to try and would be very informative.

Also, discussion and comparison with an important class of methods (*action representation learning* methods) are missing. I referred to one important work in this direction (Ref. [1]), but I'm sure there is more on this topic under different names (e.g. latent actions, action embedding learning).

---

> ### Author Response · Authors · 2021-11-10
> **Answer 1/3**
>
> We thank the reviewer for their thorough review. We hereby answer all the questions and update the paper accordingly.
> Before getting into the details, we would like to emphasize once again how hard the tasks we are tackling are (e.g. adroit environments with sparse/no reward and human demonstrations). We hope to convince you, with the additional experiments we provide, that all baselines fail to solve these tasks while AQuaDem shows very impressive results, previously unseen in the literature. What is more, the results are developed in no less than 3 setups, on 4 Adroit tasks and 9 robodesk tasks, all with human datasets.
>
> **Sec.1, paragraph 2 n°1**
>
> We apologize if  this sentence about discrete action spaces vs. continuous actions was not accurate enough and we agree that very large discrete action spaces are not more manageable than continuous ones. Yet, in our approach, the number of actions is a hyperparameter that is always chosen/found to be less than 20. Of course it depends on the task, but we can always fix it to a reasonable number, at the potential cost of degrading the learnt policy. So, what we say in the introduction holds for small action spaces, which is the context where we prove our algorithms to be useful. With discrete small action spaces, no explicit actor is needed and the computation of the max/argmax is tractable. We clarified this in the manuscript. We still argue that exploration is made easier not only by the reduction of the number of actions but also through the introduction of an exploration bias learned from human data. Combinatorial action spaces are out of scope of this paper, but interesting research direction (AQuaDem could be applied to them, given a meaningful distance in the action space). We left this for future work. The introduction has been modified to take this comment into account.
>
> **Sec.1, paragraph 2 n°2**
>
>  What we meant for (2) was not “Q-learning style” but “actor-critics”. By “tracks the maximum”, we meant there is a need for an actor and by “bootstrapped value function”: implicitly says there’s a critic. We clarified this sentence in the manuscript and hope the reviewer is happy with the new sentence.
>
> **Bang-bang controller**
>
> We agree we abused the naming of “bang-bang controller” and extended it to describe uniform discretization. We updated the manuscript to make this clearer.

---

> > ### Comment · Reviewer_MjVt · 2021-11-21
> > **Response to Authors' Rebuttal (1/3)**
> >
> > Thanks for your response. I will respond to the authors' rebuttal in three parts.
> >
> > **Sec. 1, paragraph 2:**
> >
> > This is now addressed to a reasonable extent.
> >
> > **Sec. 1, paragraph 2:**
> >
> > The fix doesn't address my point. Actor-critic methods are not limited to those that rely on the *maximum* of a Q function. E.g., PPO does not rely on MC updates (i.e. Reinforce style) and instead learns critic, without relying on maximizing a Q-function. Any algorithm that relies on such maximization is a variant of Q-learning (e.g. DDPG, TD3, SAC, etc.). The exposition remains problematic to me still.
> >
> > **Bang-bang controller:**
> >
> > The authors have added one instance of clarifying this in a super minimal fashion, by adding a short clarification in parenthesis in front of reference to the BB controller. This is helping only in the most minimal fashion but requires a more major revision of the draft to reduce the confusion usage of the terminology throughout. Though this was not a major concern, to begin with, it still remains unaddressed in my view.

---

> ### Author Response · Authors · 2021-11-10
> **Answer 2/3**
>
> **Sec. 1, paragraph 3 n°1**
>
> Indeed, this sentence was not precise enough.  Andrychowicz et al. (2020) assumes the independence between action dimensions and not a causal relationship (this was more clearly stated in the related work). We updated the manuscript to take this comment into account.
>
> **Sec. 1, paragraph 3 n°2**
>
> Action-dimensions independence is quite easy to implement indeed, although it requires changes to both policy and critic loss in SAC for example (e.g.  nt more reparameterization trick).
>
> Nonetheless, all causal relationships, like the one implemented by Metz et al. are much more complex to implement (they are autoregressive, use one MLP per action dimension). They also struggle at overperforming SAC, as demonstrated in this recent papers [2, Tavoli et al.]. Generally speaking, the variety of approaches to autoregressive density estimation [3] (which is even simpler than the “autoregressive conditional density estimation” we need for RL policies) suggests that this is quite hard to solve.
>  Policies like the one implemented by AlphaStar [1] are both complex and task specific (they are tailored for starcraft and include pointer networks, causal relations between action components etc…).
> Our approach does not require any knowledge/priors on the structure of the action space as it is solely based on demonstrations. We updated the paper to be clearer.
>
> [1] Vinyals, Oriol, et al. "Grandmaster level in StarCraft II using multi-agent reinforcement learning." Nature 575.7782 (2019): 350-354.
> [2] Vieillard, Nino, et al. "Implicitly Regularized RL with Implicit Q-Values." arXiv preprint arXiv:2108.07041 (2021).
> [3] Papamakarios, George, et al. "Normalizing flows for probabilistic modeling and inference." arXiv preprint arXiv:1912.02762 (2019).
>
> **Sec. 2, paragraphs 1-2**
>
> We indeed consider deterministic state-action reward functions, everything can be straightforwardly extended to stochastic functions, at the cost of heavier notations. We clarified this in the paper.
>
> **Sec. 4.1, paragraph 2 (last line)**
>
> What we explicitly write is that naive discretization approaches cannot be applied. We agree that these more complex approaches may be applicable, but:
> - Results on Mujoco are not competitive with SAC in terms of return (Metz et ak., Tavakoli et ak., Tang et ak.). Concerning [2,3], it is hard to tell as they do not compare to SAC, while it is SOTA, and do not consider Mujoco but other environments). If one of them was commonly accepted as the state-of-the-art for continuous control, we would be happy to include them as baselines. Yet, we argue that SAC is a state of the art algorithm for continuous control and that its counterpart SACfD is when some demonstrations are available.
> -  They do not use available data, contrary to us.
> - They do not allow the use of any discrete RL method on top of them as our approach does.
> - Our method is generic and we show how to use it in different settings in a seamless fashion. Even if it didn't outperform ad hoc algorithms in specific tasks, outperforming a standard generic baseline is what we aim for.
>
>
> **Sec. 4.3, Evaluation & results**
>
> What we say is that naive (uniform) discretization does not scale, nothing more. The aforementioned approaches would apply, but they do not make use of available human data, and for those considering a comparable setting, they’re not competitive with SAC. We thus argue we have included the strongest baselines possible. If the reviewer can provide insights on why a specific algorithm can be expected to perform better than SAC/SACfD in our setting, we would be happy to include it in the baselines.
>
> **Sec. 5, paragraph 1**
>
> Thanks, we updated the paper accordingly.
>
> **Question 1**
>
> We hope we answered all your questions with the comments above.
>
> **Question 2: Why not compare with action representation learning methods (see, e.g., Ref. [1])? This could be quite a valuable comparison, and several possibilities could be explored.**
>
> [1] experimental section does not deal with continuous control. It is specifically about reducing a huge discrete action space. AQuaDem focuses on continuous control but, with some k-means like loss, it could probably be extended to large discrete action spaces. However this is left for further work.

---

> > ### Comment · Reviewer_MjVt · 2021-11-21
> > **Response to Authors' Rebuttal (2/3)**
> >
> > ** Sec. 1, paragraph 3: **
> >
> > Partially addressed, but the additional references are not incorporated. The reference currently in place is the latest work that utilizes such a structure out of the other two references I provided. In fact, it is the least informative reference to include for the purpose of the statement.
> >
> > ** Sec. 1, paragraph 3: **
> >
> > The comparison to AlphaStar is not relevant here. AlphaStar's specificity stems from many other factors and underlying reasons. Metz et al. (2017) being more complex to implement is somewhat acceptable. Independent-style SAC being less trivial is also reasonable. However, maybe you don't need SAC. As Table 1 of Tang and Agrawal (2020) reports, Independent-style PPO with an ordinal policy already outperforms continuous SAC on Humanoid. The former also comes with a clean opensource code.
> >
> > In any case, independent or autoregressive-style agents not outperforming continuous-SAC is one thing, but mentioning in the paper that there are no discrete methods to handle such action spaces is another thing. At least, you should have made an effort to get the narrative right throughout the paper.
> >
> > *Side note (which has not affected my decision):* While I acknowledge that the following paper is recent and that it was not available on arXiv before your submission, I only wish to make the point here that if you really made an effort to consider such discretized baselines as a point of comparison, you'd likely find out how well they could work in practice (see Seyde et al. (2021) which uses Bernoulli distributions manage to outperform standard continuous-action methods). This paper also hints that maybe we don't need learned discretizations. But this is something we can't know, as you don't compare with any such baseline.
> >
> > > Tim Seyde, Igor Gilitschenski, Wilko Schwarting, Bartolomeo Stellato, Martin Riedmiller, Markus Wulfmeier, Daniela Rus. *Is Bang-Bang Control All You Need? Solving Continuous Control with Bernoulli Policies*. NeurIPS 2021.
> >
> > **Sec. 4.1, paragraph 2 (last line):**
> >
> > I fundamentally disagree with your approach to responding to this concern. The point of the paper is that utilizing *demonstrations* to *discretize* actions in a *task-relevant manner* boosts later learning on the task. Okay. What you have to show is that *unform discretization*, which *does not need demonstrations* (isn't this ideal?!) works worse. You haven't shown this. Stating that one of those papers should be a well-established SOTA to be examined in your context is not good science in my view. Such a study is highly important for evaluating the significance of your contributions, but you skip it because, e.g., "*they do not consider Mujoco but other environments*". I strongly believe that you should try some of these methods, tune them with the similar effort you made for evaluating your approach, and test them in a unified evaluation scheme as your methods to know for sure.
> >
> > **Sec. 4.3, Evaluation & results:**
> >
> > Again, same issues:
> > - Not making use of human data is not a disadvantage vs. using it in my view. If naive discretization works better without human data vs. informed discretization using human data, then that's better for me.
> > - Naive discretization not working as well as continuous counterparts: you can't claim this, as you didn't try.
> > - "*What we say is that naive (uniform) discretization does not scale, nothing more.*": And what I say is that they do (as in the referenced papers). Scaling but not performing as well as SAC would be a different statement.
> >
> > **Sec. 5, paragraph 1:**
> >
> > The first two points were incorporated. Is there a reason why the references I pointed out are not included in the revised version?

---

> ### Author Response · Authors · 2021-11-10
> **Answer 3/3**
>
> **Question 3: Why only try the discrete-action baseline in the low-dimensional case of the Robodesk domain? As you have noted in the related work, there are now a few approaches for scaling discrete-action Q-based and PG methods to high dimensional problems; why not try them?**
>
> We do not try them because they report performance much below the ones of SAC (e.g. Tang & Agrawal), and they do not make the most of demonstrations. We don’t claim to provide a benchmark on discretization methods but simply to introduce a method that uses demonstrations to discretize the action space and use a discrete deep RL algorithm to solve continuous control tasks.
> We would like to take this opportunity to engage in a discussion with the reviewer about the best baseline in their opinion. We chose SAC and SACfD on purpose after serious considerations because 1/ it's SotA performance, 2/ its ability to handle demonstrations in a principled but easy way. Yet we are happy to hear arguments in favor of another baseline that would support our claims: handling demonstrations, genericity (3 different setups), strongly performing in continuous-control environments etc.
>
> **We start with Question 6 which will help answer question 4: Choosing hyperparameters that perform best on average across all tasks (including test ones) is not fair. Am I misunderstanding something here, or is there a reason why this makes sense here?**
>
> The notion of train/test environments was defined for Atari to tackle the high computation power needed to run these 60 games. Yet it is not common at all in RL. Most papers tune hyperparameters separately for each environment! What we provide here is the performance of a single configuration across different environments.
>
> We are confident that our hyperparameter selection is fair. It follows the same principle for all baselines and AQuaDem. We do not report the results of best performing configuration of the sweeps, but rather select greedily the value that leads to the best performance for each hyperparameter, and then run again, for 10 seeds, the configuration created this way. More details can be found in the appendix (and supported by the histograms in the html files in the supplementary material). But please note that this hyperparameter selection process follows high standards in RL.
>
> **Question 4: How does the tuning process on Relocate work?**
>
> The process is the one described beforehand but taking performance of relocate instead of the average over tasks. Note that the process was followed for our algorithm, as well as all the baselines! We did this specifically to showcase that, even for the hardest environment of the task suite (relocate requires generalization of initial position and target position of the ball, as one can see in the videos we provide in supplementary material), AQuaDQN was still able to perform suprisingly well (assuming it is tuned for this environment specifically, but a single change was required as told in the last sentence of  D.4.1).
>
> **Question 5**
>
> The success is defined as a 0/1 boolean for each episode. The adroit tasks have fixed length episodes (of 200 steps), and the sparse reward is 1 if the task is solved and 0 otherwise. If the agent solves the task after 150 steps, it will then receive 1 at each steps starting at step 150, which will lead to a sparse return of 50 (assuming the agent does not “unsolve” the task, for example closing the door again).
>
> **Question 7: Could you position your work w.r.t. the line of work of action-embedding/action-representation/latent-action learners (e.g. in the context of Refs. [1,5])?**
> We are happy to enhance our related work. Yet as we mentioned, the very specific feature of our work is to leverage human data which distinguishes it from this line of work. Also they mostly focus on hybrid setups and do not outperform SAC on non-hybrid continuous-control tasks.
>
> Thanks again for the thorough review.

---

> > ### Comment · Reviewer_MjVt · 2021-11-21
> > **Response to Authors' Rebuttal (3/3)**
> >
> > Fair responses are provided to questions 4, 5, and 6. But regarding question 3, my concern remains unresolved (see my response part 2/3 for related discussions about the issue). Regarding question 7: fine; I don't agree that your work should not consider approaches that do not utilize human demos (as I have previously argued in part 2/3 of my response), but Ref. [1] is not critical for me to include as it does not discuss action discretization directly. But it would be useful if you discuss it and find some deep connections.

---

> > > ### Author Response · Authors · 2021-11-22
> > > **Response to the Reviewer**
> > >
> > > We thank the reviewer for engaging in a helpful discussion.
> > >
> > >
> > > ### Minor edits:
> > > - We have incorporated the missing related works.
> > > - We changed the wording introducing BB-3 and BB-5 to make it more speaking.
> > > - Section 2 paragraph 2: We have replaced the term “tracks a maximum”  by “optimizes” which we believe is more general and encompasses both Q-learning style methods and on-policy methods based on bootstrapped estimates. Regarding PPO, indeed it does not optimize for the maximum Q-value, because of the clipping that heuristically replaces the KL constraint of TRPO, which can be seen as a soft-maximum. We think that the term “optimize” is thus well-adapted.
> > >
> > >
> > > ### Adjusting the naive discretization narrative:
> > > We believe that the reviewer will agree that the naive discretization method does not scale with the action dimensionality without additional assumptions e.g. action dimension independence or autoregressive representation (by naive, we really mean discretizing all dimensions and considering all possible combinations). We added the precision in the revised version (last line of paragraph 2 in Section 4.1). The following sentences in the related work also explain our point: “this naive discretization is problematic in high-dimensional action spaces, as the number of actions grows exponentially with the action dimensionality. To mitigate this phenomenon, a possible strategy is to assume that action dimensions are independent [...]”.
> > >
> > >
> > > ### On adding new baselines:
> > > We considered what we thought were the strongest baselines regardless of whether it was based on a discretization procedure (SAC and SAC+demonstrations). We tuned the baselines with the same amount of compute that we used for the proposed method. We evaluated them on 10 seeds and provided videos for *all* resulting agents. We argue that the results we provided, together with the thorough evaluation protocol, already empirically validate our method.
> > > Nevertheless, we agree with the reviewer that the comparison to discretization method baselines could be included at the very least for the sake of completeness. We will include the method pointed out by the reviewer (Seyde et al. 2021), which actually appears competitive with SAC, in the camera-ready version. We won’t have time to run these experiments by the revision deadline, which is today; yet, given the results provided by Seyde et al, we do not expect this to change our results: quoting their Fig 2, “Generally, we find that the Bang-Bang policies perform on par with the normal Gaussian policies.”
> > >
> > > Again, we thank the reviewer for helping improve the submission and the mostly positive appreciation of our work. Please let us know if you have remaining concerns.

---

> > > > ### Comment · Reviewer_MjVt · 2021-11-22
> > > > **Response to the Author(s)**
> > > >
> > > > Thanks for your response. I feel that a few things got overlooked here again, e.g. the fact that a discrete-PPO variant from Tang and Agrawal (2020) does indeed outperform continuous-SAC on the complex Humanoid task. The narrative of the paper looks better than before now, but I encourage the authors to *more generally* soften their statements about their contributions much more than they have so far. I personally would appreciate *true* even if *much less bold* statements, and I think that approach would have made my score an 8. But right now, even in authors' responses to me and other reviewers, I see few instances of overclaiming (I have previously pointed these out in my response to authors).
> > > >
> > > > I get a feeling that the authors instead of taking some time in their original response to really incorporate the suggestions, rushed to respond quickly and thus really didn't look closely at the comments. Regardless, I like the paper and would also invite the other reviewers to consider *marginally accepting* this paper as well.

---

> > > > > ### Author Response · Authors · 2021-11-22
> > > > > **Response to the Reviewer**
> > > > >
> > > > > We thank the reviewer for their quick response.
> > > > >
> > > > > Maybe one of the overclaims that the reviewer is referring to is the SOTA performance of SAC and SAC + demonstrations. Actually our implementation of SAC (for which we reproduced the original implementation in JAX) and for which we provided training curves in the Appendix E, strongly outperforms the results provided by Tang and Agarwal, 2020. They report performance at 6000 for SAC and 6018&pm;239 for their method for 10^7 environment steps while we have performance close to 8700&pm;250 for Humanoid for the same number of interactions. The reason for the discrepancy between performance is that the V2 version of SAC, based on adaptive temperature, performs much better than the V1 version with a fixed temperature.
> > > > >
> > > > > As requested by the reviewer and by reviewer zgiZ, we also toned down the claims made on exploration in discrete action spaces vs exploration in continuous action spaces in the manuscript.
> > > > >
> > > > > If the reviewer has another specific overclaim in mind that we have not addressed, we would happily make the adequate change. Again, we thank the reviewer for the positive recommendation.

---

> ### Author Response · Authors · 2021-11-17
> **Message to reviewer**
>
> Dear reviewer, there're only 5 days left before the end of discussion, and no interaction so far. We would be more than happy to know if our rebuttal and revision addressed your questions and comments, or, if not, what are the remaining concerns you may have.
>
> Best,
>
> The authors

---

### Official Review · Reviewer_zgiZ · 2021-11-07

**Correctness:** 3
**Technical Novelty And Significance:** 3
**Empirical Novelty And Significance:** 3
**Recommendation:** 5
**Confidence:** 4

**Main Review:**

The problem setting is interesting, timely, and could have a good amount of impact should the technique be adopted. Please consider the following comments;

- The authors “reproduced” the SAC implementation. I am not fully convinced that the implementation is competitive to the original. While running a HP sweep is commendable, it is not enough to ensure that the baselines work as intended, especially in exploration tasks that can have a tremendous impact. One would need to (a) ensure that the implementation can reproduce the results in the original paper in the original environments (mostly Mujoco/Dm control suite/openai gym) and (b) run a number of experiments of Aquagail in these environments to compare performance. While I agree that it might be harder to generate meaningful play data, I do think that this shouldn’t be a problem for the imitation learning experiments.

- Related work on exploration in continuous action spaces is largely missing. While exploration is not easy in this domain, there exists a large body of literature (all the different types of curiosity mechanisms, UCB inspired methods etc.) [2-6] If the approach claims superiority of discrete action spaces over continuous ones, a baseline with principled exploration in continuous action spaces is necessary.

- Why not compare the approach against a hierarchical Rl/Imitation learning approach or a hybrid continuous-discrete baseline? E.g. [1] The authors also claim that this is the most related approach.

- The approach states: “Our objective is to reduce a continuous control problem to a discrete action one, on which we can apply discrete-action RL methods.” I believe this alone is already a great contribution and it is not necessary to frame it as an advantage over continuous action-space RL. The paper makes a number of somewhat handwavy claims why discrete action spaces are superior but while they may sound intuitive they are not substantiated by theory or experiments.

- How does the approach compare to state-of-the-art in offline RL, which provides a similar problem setting?

- 10 seeds is amazing and I would love to see more papers following this direction. Nonetheless, it is not clear what the shaded regions in the plots mean. Is this the 1 standard deviation interval or a confidence interval based on the t-test? What p-value of the confidence interval is chosen? I agree that this is a minor detail point but it should be mentioned.


[1] Neunert, M., Abdolmaleki, A., Wulfmeier, M., Lampe, T., Springenberg, T., Hafner, R., ... & Riedmiller, M. (2020, May). Continuous-discrete reinforcement learning for hybrid control in robotics. In Conference on Robot Learning (pp. 735-751). PMLR.

[2] Osband, I., Blundell, C., Pritzel, A., & Van Roy, B. (2016). Deep exploration via bootstrapped DQN. Advances in neural information processing systems, 29, 4026-4034.

[3] Osband, I., Van Roy, B., Russo, D. J., & Wen, Z. (2019). Deep Exploration via Randomized Value Functions. J. Mach. Learn. Res., 20(124), 1-62.

[4] Chen, R. Y., Sidor, S., Abbeel, P., & Schulman, J. (2017). UCB exploration via Q-ensembles. arXiv preprint arXiv:1706.01502.

[5] Seyde, T., Schwarting, W., Karaman, S., & Rus, D. (2020). Learning to Plan Optimistically: Uncertainty-Guided Deep Exploration via Latent Model Ensembles. 5th Conference on Robot Learning (CoRL)

[6] Bai, C., Wang, L., Han, L., Hao, J., Garg, A., Liu, P., & Wang, Z. (2021, July). Principled exploration via optimistic bootstrapping and backward induction. In International Conference on Machine Learning (pp. 577-587). PMLR.

[7] Pathak, D., Agrawal, P., Efros, A. A., & Darrell, T. (2017, July). Curiosity-driven exploration by self-supervised prediction. In International conference on machine learning (pp. 2778-2787). PMLR.


**Summary Of The Paper:**

The paper proposes learning a state-dependent discretization of a continuous action space using demonstrations. The motivation is to enable the use of discrete-action deep RL methods instead of relying on policies with continuous action space parametrization. The authors argue that this results in improved exploration (avoiding the curse of dimensionality). The approach is evaluated on 5 environments in the Adroit and Robodesk environments on Reinforcement Learning with demonstrations, Reinforcement Learning with play data, and Imitation Learning. The authors claim that they outperform state-of-the-art continuous control methods both in terms of sample efficiency and performance on every setup.

**Summary Of The Review:**

The paper presents an interesting framework for learning the discretization of action spaces. Related work needs to be extended to include exploration, the claims of benefits of discrete over continuous action spaces have to be substantiated, the evaluation has to be done on fair baselines and commonly used environments to allow the comparability. I am happy to adjust my rating should these issues be addressed.

---

> ### Author Response · Authors · 2021-11-10
> **Answer 1/2**
>
> We thank the reviewer for their thorough review. We hereby answer all the questions and update the paper accordingly.
> Before getting into the details, we would like to emphasize once again how hard the tasks we are tackling are. We solve, for the first time, tasks that are notoriously hard, where most standard RL does not work and were introduced because they are more representative of real robotic tasks (e.g. adroit hand manipulation environments with sparse/no reward and human demonstrations). We hope to convince you, with the additional experiments we provide, that all baselines fail to solve these tasks while AQuaDem shows very impressive results, previously unseen in the literature. What is more, the results are developed in no less than 3 setups, on 4 Adroit tasks and 9 robodesk tasks, all with human datasets.
>
> **SAC Implementation**
>
> Our SAC implementation has been extensively benchmarked and is extremely competitive. To support this claim, we provide in Appendix E: “Sanity check Baselines” the following additional experiments:
>
> -  SAC with original paper (& adaptive temperature) default parameters on 20 millions interactions and 10 random seeds on HalfCheetah-v2, Hopper-v2, Ant-v2, Walker2d-v2, Humanoid-v2. Please note that results are consistent with what is published in the literature.
> - We also provide AQuaDQN and SAC results on these environments (all but Humanoid as D4RL does not provide a dataset for this one) with the medium-expert datasets from D4RL, using the configurations we found to be best for our experiments on Adroit (and without re-tuning them for Mujoco). Please note that, while this result is a good sanity check, it’s not so interesting as such since SAC solves these tasks very well. The purpose of this paper is to address much more complex tasks, that cannot be properly solved from scratch in the continuous domain and to use AQuaDem to make the most of human demonstrations/play data (which cannot be produced for these environments, as a human cannot control properly a environment with >17 DoF/actuators). This is why we focused on  Adroit/Robodesk manipulation tasks that are notoriously harder than the Mujoco locomotion tasks.
> Concerning the hyperparameters selection process, a fairly important amount of detail can be found in the appendix and the full results of our sweeps are in the supplementary material. We were extremely careful to put the same amount of effort tuning the baselines and our algorithms.
>
> **Related work on exploration**
>
> We would like to highlight that our approach is based on human demonstrations which, in some sense, also addresses the exploration issue. While exploration in continuous action spaces is indeed a hard problem, we don’t think we should address it in a standard way here and compare exploration in the reduced discrete action space with exploration in the original continuous action space. Indeed, our claim about exploration is mostly that thanks to their priors about the task, humans already solved a large part of the exploration problem in the continuous action space. Thus the agent’s exploration is biased to follow the human’s prior.  As such, it is close to approaches such as DQfD or DDPGfD/SACfD in the continuous action setting, that we discuss/consider as baselines. We probably didn’t make ourselves clear enough in our intro about this, and we will clarify in a revised version.
> Yet, we want to address the reviewer’s concerns for the sake of completeness. Please note the following: SAC is state-of-the art on continuous-action reinforcement learning problems. None of the approaches in [2-6] beat SAC. If they were, they would provide experiments in continuous control domains to prove so. We would consider them as the new SOTA and natural baselines. If the reviewer thinks a specific algorithm outperforms SAC in our setting, we would be happy to add them as baseline, but we would like to see evidence that it is the case (like a paper/blogpost showing this algorithm beats SAC/TD3 in a closely related setup). [2] only test on Atari (discrete actions), [3] is mostly theoretical and does not provide results in continuous control, [4] only provides results on Atari, [5] fails to demonstrate results outperforming SAC, [6] only tests on Atari.
>
> **Related work on HRL**
>
> [1] considers a hybrid setup (both discrete and continuous actions) which is not the setup considered here. We believe that AQuaDem is not limited to continuous action spaces but could be extended to hybrid ones. As this would be an extension of our work, we don’t think it could serve as an appropriate baseline for the considered setting.

---

> ### Author Response · Authors · 2021-11-10
> **Answer 2/2**
>
> **Questions on claims**
>
> Indeed, the goal is to reduce continuous control to discrete control using a dataset of human demonstrations, which means meaningful interactions. This is key to our approach and somehow missed by the reviewer’s sentence. In this context (where we use *human* interactions to discretize), our experiments show that it’s better than continuous control (e.g., vs SACfD in RL+demo, vs SAC in RL+play), so we argue it is actually substantiated by experiments.
> Yet, we understand what the reviewer means by saying some claims in the introduction are hand-wavy, we take note of it and update the introduction to tone down these claims.
>
> **Comparison to Offline RL**
>
> Contrary to offline RL, we allow for further interactions  with the environment to train the agent. The dataset is only used to train the action discretization scheme.We chose 3 different online settings on purpose to show the genericity of the method when it comes to learning in interaction with the environment.
>
> **Confidence interval**
>
> As already specified in each caption of each figure, it is the interquartile range (75th percentile - 25th percentile https://en.wikipedia.org/wiki/Interquartile_range).
>
> **Answer to Summary**
>
> -  On including exploration baselines: only 1 of  our 3 setups have sparse rewards. ⅔ of our experiments on imitation learning and play data+dense reward have denser rewards. We think we provided a quite good range of diversity in this respect. What is more, we hope we were convincing enough when making  the argument above that SAC and SACfD are the fairest baselines one could design for these experiments, and that they were tuned with extreme care (please check the appendix and the supplementary material for additional details).
> - Claims have to be substantiated (on discrete vs continuous):
>      - We are happy to tone-down and clarify generic claims in the introduction.
>      - We insist that we focus on using human demonstrations which results in strong performance as it is substantiated in our experiments.
>      - What is more, we are transparent on the limitations of our approach (not outperforming BB-2 in RoboDesk, ablation in Section 3).
> - Fair baselines:
>     - We strongly think that our baselines are fair, we consider SOTA baselines whenever possible in all settings using the same data, same number of interactions, same HP tuning sweeps.
>     - We’re open to reviewer suggestions if we missed some baselines, but we would appreciate to have a better understanding of how the suggested baselines could be implemented and what are the evidences that  it could outperform the current one (SAC, SACfD).
> - Commonly used experiments:
>      - Unfortunately, commonly used environments such as Mujoco are too simple to demonstrate the value of AQuaDem and cannot be controlled by humans. We provided experiments of AquaDQN to  show extremely strong performance on a very hard setup. As we mentioned, we are still happy to provide the results of our SAC on the Mujoco environment to prove it is very strong. We remind that the environments were chosen to allow having human demonstrations (the core subject of this paper). Adroit and Robodesk are now quite common environments too.
>
>
> Thanks again for the numerous comments, we would be happy to interact more with you if necessary and are totally open to suggestions.

---

> ### Author Response · Authors · 2021-11-17
> **Message to reviewer**
>
> Dear reviewer, there're only 5 days left before the end of discussion, and no interaction so far. We would be more than happy to know if our rebuttal and revision addressed your questions and comments, or, if not, what are the remaining concerns you may have.
>
> Best,
>
> The authors

---

### Official Review · Reviewer_teiU · 2021-11-16

**Correctness:** 3
**Technical Novelty And Significance:** 3
**Empirical Novelty And Significance:** 3
**Recommendation:** 5
**Confidence:** 4

**Main Review:**

This paper explores the connection between the continuous actions space and discrete actions space, proposes an automatic discretization. This idea is interesting, the writing is clear and easy to follow. But I have a few concerns on this paper:

(1) The benefit of proposed method relies on the preference of discrete action space over the continuous action space, as it mainly provides an architecture to output multiple continuous actions and evaluate them with discrete action based Q-learning methods. However, this preference is not discussed fully in the paper. The most relevant discussion in the paper is practical comparison between proposed approach and SAC, a continuous action space based baseline. Though the practical performance of the proposed discretization looks better, it is not clear if this advantage is task specific. More analysis on this part could better support the motivation of this paper.

(2) The role of the temperature T of equation (1) is not very clear. The paper states that this hyperparameter controls whether all output actions are similar to the demonstrated action or only one of them is similar to the demonstrated action. But it is not convincing enough. The proposed loss function sums up losses from all output K actions, and the temperature T is the same for all actions and hence does not seem to control significance of any individual action. As a result, the temperature seems to only control the scale of the loss.

(3) Another concern is also related to the proposed loss. The proposed loss does not seem to provide guidance on the diversity of output actions. As they are bound to the same demonstrated action, it is difficult to tell how it can output multiple different actions for the same input. If the model always output similar actions in each candidates for the same input, the significance of the proposed design can be strongly weakened. Because the proposed loss is exactly the ordinary BC when K = 1, as the paper mentions.

(4) As the supplementary (A) section shows, the proposed architecture looks very similar to the Gaussian mixture models. However, the paper does not compare ordinary Gaussian mixture models with the proposed method. The usage of Guassian mixture models in reinforcement learning and imitation learning problem does not seem to be novel, as we can find it in some previous works:

Chernova, Sonia, and Manuela Veloso. "Confidence-based policy learning from demonstration using gaussian mixture models." Proceedings of the 6th international joint conference on Autonomous agents and multiagent systems. 2007.

Agostini, Alejandro, and Enric Celaya. "Reinforcement learning with a Gaussian mixture model." The 2010 International Joint Conference on Neural Networks (IJCNN). IEEE, 2010.

Choi, Yunho, Kyungjae Lee, and Songhwai Oh. "Distributional deep reinforcement learning with a mixture of gaussians." 2019 International Conference on Robotics and Automation (ICRA). IEEE, 2019.

Then the novelty of proposed method may also strongly depends on how it is better than related previous works. The novelty and significance of this paper can be shown clearer if convincing analysis and comparison to previous Gaussian mixture models works are provided.

**Summary Of The Paper:**

This paper looks into improving data efficiency in continuous action space reinforcement learning problem by connecting it to its discrete action space correspondence and taking advantage of demonstration data. To be specific, the paper proposes learning a state dependent discretization of the original continuous action space by mimicking actions in demonstration trajectories. This discretization output a fixed number of actions for each input state. In a grid world example, it is visualized that the proposed method learns multimodal demonstrated action better as it can catch different actions in the same state. In the experiment section, the proposed method is compared to baselines in various demonstration and reinforcement learning setting and shows benefit over compared baselines.

**Summary Of The Review:**

The paper provides an interesting idea on connecting discrete action space and continuous action space for reinforcement learning and imitation learning. The writing is clear. But some important discussion seems to be insufficient and hence its novelty and significance is not clear enough.

---

> ### Author Response · Authors · 2021-11-17
> **Answer to reviewer 1/2**
>
> (1)
>
> The contribution of this work is two-fold and seems somewhat missed by the reviewer’s comment. First, this work shows how to make the most of **human** demonstrations (which are non-markovian, non-stationary, non-deterministic…) to learn a good policy. Once again, you can check in the videos the remarkable capabilities of AQuaDem to learn policies very similar to the ones of humans, whilst baselines fail at doing so. Second, the discretization scheme we propose unlocks the out-of-the-box usage of discrete RL methods on continuous-control problems. We discuss in the introduction the advantages this brings:
> An easier exploration, because it relies on the human priors (even when the human is not solving the task at hand, as demonstrated in the Play Data experiments).
>
> Second, the exact computation of the greedy policy which is not possible otherwise. This has various consequences, but typically, it allows one to use efficient regularization techniques like Munchausen (https://proceedings.neurips.cc//paper/2020/file/2c6a0bae0f071cbbf0bb3d5b11d90a82-Paper.pdf) which is not possible in the continuous control setting.
>
> (2)
> The role of temperature T is quite detailed in section 3, here we provide additional mathematical insights. Basically, the loss function of Aquadem is a scaled and signed log-sum-exp (or LSE for short), that is a soft-minimum. To see this, just consider the unsigned scaled LSE (relying on the fact that $\min(x_1…x_k) = - \max(-x_1 … -x_k)$. The scaled LSE is the convex conjugate of the negative Shanon entropy (scaled by the temperature T), that is the maximum of the associated Legendre-Fenchel transform (while the softmax is the maximizer). From this, we can directly see that the LSE approximates the maximum for low temperature (the softmax approximates the argmax), while it approximates up to a constant the average for high temperatures (the softmax approximates a uniform distribution). To make things even clearer, we provide below a self-contained and simple proof (not relying on convex conjugacy) in appendix F.
>
>
> (3)
>
> Here again, we think that the reviewer missed the fact that  our goal was to make the most of human demonstrations (note that **all** our experiments are done with human-collected data, except the one added to the revision in the appendix to address a reviewer’s comment). By essence, our work was proposed to model this multimodal/non-markovian data. If the demonstrator is purely unimodal and markovian, then indeed, our method won’t do better than BC, but this is an extremely strong assumption, and the results of our experiments, where we compare to BC, proves this undoubtedly (results are much better than BC on the human data).

---

> ### Author Response · Authors · 2021-11-17
> **Answer to reviewer 2/2**
>
> (4)
> Once again, our contributions are to unlock the usage of discrete RL methods and to solve human-data augmented RL problems. This is totally orthogonal to the usage of GMM as a policy. The idea of discretizing actions using human data and then relying on discrete action RL to solve continuous control problems is highly novel and demonstrating its power in three different setups is, in our humble opinion, a fairly important contribution. We read the papers that the reviewers mention.
>
> -  "Confidence-based policy learning from demonstration using gaussian mixture models":
> This paper learns a GMM policy by querying an expert, without doing RL. The setting is closer to the one of Dagger (https://www.ri.cmu.edu/pub_files/2011/4/Ross-AISTATS11-NoRegret.pdf), which is quite orthogonal to ours. As we do not allow querying the expert, this would correspond to a BC baseline, which we already included. Regarding using a GMM as a policy in an RL agent (not covered by the paper pointed by the reviewer, though), notice that we consider the strong SAC baseline, and that the former version of SAC was using a GMM (see ICLR 2018 version, sec 4.3, https://openreview.net/forum?id=HJjvxl-Cb), with less good results than the later version we reproduce (we think notably because GMMs do not allows using the reparameterization trick)
>
> - "Reinforcement learning with a Gaussian mixture model": Here the GMM is used to learn the Q-function, so this is again orthogonal to our contribution. This paper experiments with the Cartpole environments while we provide results in complex hand manipulation tasks which are harder of magnitude more complex.
>
>  - "Distributional deep reinforcement learning with a mixture of gaussians". The GMM is here used for modeling the random return (as done in distributional reinforcement learning), which is again not relevant in our context
>
> To sum up, beyond the common keywords “GMM” and “RL”, we hardly see how these works are relevant in the context of our contribution, and even more why they would be meaningful baselines or how they could question the novelty of our work. We would appreciate it if the reviewer could expand on this, in case we missed something.
> We are very clear on the connections to GMM in the paper (as appendix A demonstrates). But the core contribution is **not the loss itself** but **its use to discretize a continuous control action space** in a **state dependent** manner using **human demonstrations** and the experimental demonstration that this **works surprisingly well**.

---

### Author Response · Authors · 2021-11-10
**General answer to reviewers**

We thank the reviewers for their review. We emphasize a few points before summarizing the changes made to the manuscript.

1. We solve, for the first time, tasks that are notoriously hard, where most standard RL does not work and that were introduced because they are more representative of real robotic tasks (e.g. adroit hand manipulation environments with sparse/no reward and human demonstrations). We hope to convince, with the additional experiments we provide, that all baselines fail to solve these tasks while AQuaDem shows very impressive results, previously unseen in the literature. What is more, the results are developed in no less than 3 setups, on 4 Adroit tasks and 9 robodesk tasks, all with human datasets. We provide results in terms and different metrics and show the learned behaviors in several videos provided in the supplementary material. We hope this is enough to showcase this simple yet powerful idea of using a discrete deep RL method on top of learned action candidates can be useful in many scenarios.

2. We argue this approach is totally novel and showcasing that it can work is already an important contribution (yet we still worked hard to provide fair baselines in the setup).

3. The work focuses on the usage of human data. It is not about discretization but about discretization from demonstrations.

4. We thoroughly reviewed literature and chose SAC and SACfD on purpose after serious considerations because 1/ it's SotA performance on continuous control tasks 2/ it’s able to handle demonstrations in a principled but easy way. Yet we are happy to hear arguments in favor of another baseline that would support our claims: handling demonstrations, versatility (RL with sparse/dense reward, imitation), strongly performing in continuous-control environments etc.

**We made the following changes to the manuscript:**

- We toned-down the “hand-wavy” claims in the introduction and made more specific claims about small discrete action spaces that, we argue, are backed by our experiments in the rest of the papers.
- We took all reviewers suggestions into account to enhance the introduction and the related work.
- We added to the appendix (Appendix E: Sanity check baselines) the experiments suggested by reviewer zgiZ, demonstrating the strong performance of our baselines.
- We made all the minor changes asked by the reviewers.

We hope this, along with our thorough answers  to their specific questions, will make things clearer. We are happy to keep improving the paper if necessary and answer more questions.

---

> ### Author Response · Authors · 2021-11-18
> **2nd revision**
>
> Dear reviewers,
>
> We updated the paper to include a proof concerning the influence of the temperature to answer reviewer teiU.
>
> We poster most our answers more than a week ago and would really be more than happy to know if our rebuttal and revision addressed your questions and comments, or, if not, what are the remaining concerns you may have.
>
>
> Thanks you again.

---

### Public Comment · ~Yali_Du1 · 2021-11-16
**Clarifications on notations**

Hi, thanks for the interesting work. There are some notations that I found confusing. Hope the authors will help to clarify this.

1- The calligraphic symbol $\mathcal{A}$, is it representing a continuous action space or a discrete action set?

2- if $\mathcal{A}$ indicates a discrete action set, how could you compute the distance of $\Psi_k(s)$ and $a$ in Eq.(1)?

If $\mathcal{A}$ indicates a continuous action set, then the output $a_k$ of $\Psi_k(s)$ is continuous action. how could $a_k$ be passed into a discrete Q-function?

---

> ### Author Response · Authors · 2021-11-16
> **Re: Clarifications on notations**
>
> Hello! Thanks a lot for the interest and for the question !
>
> In Section 2 "Preliminaries", the MDP description is generic and $\mathcal{A}$ can designate both a continuous or a discrete action space. Your question problably concerns more the Section 3 "Method".
>
> In this part, $\mathcal{A}$ designates the continuous action space. Yet your question is totally relevant because there is a slight abuse of notation in the figure: when we write $Q(s, a_k)$ what we mean is $Q(s)_k$ (note that it is mathematically correct but could indeed lead to confusion). The discrete RL agent that runs on top of the candidates actions does not know what $a_k$ is. It just takes action $k$ (corresponding to the greedy $\max_k Q(s)_k$ where $Q(s)$ is the vector of logits). It is the orginial DQN algorithm that runs, and it is not "aware" that the meaning of action $k$ changes over state space.
>
> Yet, note that an interesting variation of our algorithms could be to actually feed the action candidates to the Q network (as done for the critic network in SAC or TD3). We left this for future work as we deemed very interesting the fact that you could use DQN/DQfD/DQNwithGAILreward on top of the AQuaDem candidate actions without changing the original algorithms in any way ! And they would solve the tasks very efficitiently !
>
> We will update the manuscript to make this clearer.
>
> Thanks again for the question.

---

> > ### Public Comment · ~Yali_Du1 · 2021-11-17
> > **Followup question**
> >
> > If I understand correctly, $\mathcal{A}$ designates the continuous action space, and action $k$ is only a symbol without knowing $a_k$.
> >
> > If so, in the figure in Page 3, how can you link the $a_3$ predicted by $\Psi_k(s)$ with $k$-th action $a_3$ in step 2?

---

> > > ### Author Response · Authors · 2021-11-17
> > > **Re: Followup question**
> > >
> > > Thanks for the followup, sorry if it is still unclear!
> > >
> > > $k$ is not just a symbol! For a given state DQN outputs $K$ logits (1 for each action candidates). If DQN selects action $k$, $a_k$ (which is state dependent) is played. DQN will then update its belief on Q-value associated to action $k$.
> > > Said otherwise, DQN is run out-of-the-box with K actions, and when action $k$ is selected in state $s$, then $a_k(s)$ is played.

---

### Decision · Program_Chairs · 2022-01-20

**Decision:**

Reject

**Comment:**

The paper presents a reinforcement learning technique for problems with continuous actions. The proposed approach consists in learning  a discretization of continuous action spaces from human demonstrations. This discretization returns a fixed number of actions for each input state. By discretizing the action space, any discrete action deep RL technique can be readily applied to the continuous control problem. Experiments reported in the paper show that the proposed approach outperforms several RL baselines such as SAC.

The key criticism from the reviewers relates to the incremental nature of this paper's contribution. While the precise equation proposed by in this paper for learning discrete actions from demonstrations may be novel, there have been several very similar techniques in the literature. For example, Gaussian Mixture Models (GMMs), a closely related model, have been widely studied in the context of learning policies from demonstrations.

In summary, the reviewers are not convinced that the paper contains sufficiently novel ideas for an ICLR publication.